# Decentralized SGD with Asynchronous, Local, and Quantized Updates

## Abstract

The ability to scale distributed optimization to large node counts has been one of the main enablers of recent progress in machine learning. To this end, several techniques have been explored, such as asynchronous, decentralized, or quantized communication–which significantly reduce the cost of synchronization, and the ability for nodes to perform several local model updates before communicating–which reduces the frequency of synchronization.

In this paper, we show that these techniques, which have so far been considered independently, can be jointly leveraged to minimize distribution cost for training neural network models via stochastic gradient descent (SGD). We consider a setting with minimal coordination: we have a large number of nodes on a communication graph, each with a local subset of data, performing independent SGD updates onto their local models. After some number of local updates, each node chooses an interaction partner uniformly at random from its neighbors, and averages a possibly quantized version of its local model with the neighbor's model. Our first contribution is in proving that, even under such a relaxed setting, SGD can still be guaranteed to converge under standard assumptions. The proof is based on a new connection with parallel load-balancing processes, and improves existing techniques by jointly handling decentralization, asynchrony, quantization, and local updates, and by bounding their impact. On the practical side, we implement variants of our algorithm and deploy them onto distributed environments, and show that they can successfully converge and scale for large-scale image classification and translation tasks, matching or even slightly improving the accuracy of previous methods.

## 1 Introduction

Several techniques have been recently explored for scaling the distributed training of machine learning models, such as communication-reduction, asynchronous updates, or decentralized execution. For background, consider the classical *data-parallel* distribution strategy for SGD (Bottou, 2010), with the goal of solving a standard empirical risk minimization problem. Specifically, we have a set of samples $S$, and wish to minimize the $d$-dimensional function $f : \mathbb{R}^d \to \mathbb{R}$, which is the average of losses over samples from $S$, by finding $x^\star = \text{argmin}_x \sum_{s \in S} \ell_s(x)/|S|$. We have $n$ compute nodes which can process samples in parallel. In data-parallel SGD, each node computes the gradient for one sample, followed by a gradient exchange. Globally, this leads to the iteration:

$$x_{t+1} = x_t - \eta_t \sum_{i=1}^{n} \tilde{g}_t^i(x_t),$$

where $x_t$ is the value of the global parameter, initially $0^d$, $\eta_t$ is the learning rate, and $\tilde{g}_t^i(x_t)$ is the stochastic gradient with respect to the parameter $x_t$, computed by node $i$ at time $t$.

When executing this procedure at large scale, two major bottlenecks are *communication*, that is, the number of bits transmitted by each node, and *synchronization*, i.e., the fact that nodes need to wait for each other in order to progress to the next iteration. Specifically, to maintain a consistent view of the parameter $x_t$ above, the nodes need to broadcast and receive all gradients, and need to synchronize globally at the end of every iteration. Significant work has been dedicated to removing these two barriers. In particular, there has been progress on *communication-reduced* variants of SGD, which propose various gradient compression schemes (Seide et al., 2014; Strom, 2015; Alistarh et al., 2017; Wen et al., 2017; Aji and Heafield, 2017; Dryden et al., 2016; Grubic et al., 2018; Davies et al., 2020), *asynchronous* variants, which relax the strict iteration-by-iteration synchronization (Recht et al., 2011; Sa et al., 2015; Duchi et al., 2015), as well as *large-batch* or *periodic model averaging* methods, which aim to reduce the *frequency* of communication (Goyal et al., 2017; You

et al., 2017) and (Chen and Huo, 2016; Stich, 2018), or *decentralized* variants, which allow each node to maintain its own, possibly inconsistent, model variant (Lian et al., 2017; Tang et al., 2018; Koloskova et al., 2019). (We refer the reader to the recent surveys of (Ben-Nun and Hoefler, 2019; Liu and Zhang, 2020) for a detailed discussion.) Using such techniques, it is possible to scale SGD, even for complex objectives such as the training of deep neural networks. However, for modern large-scale models, the communication and synchronization requirements of these parallel variants of SGD can still be burdensome.

**Contribution.** In this paper, we take a further step towards removing these scalability barriers, showing that all the previous scaling techniques—decentralization, quantization, asynchrony, and local steps—can in fact be used *in conjunction*. We consider a highly decoupled setting with $n$ compute agents, located at vertices of a connected communication graph, each of which can execute sequential SGD on its own local model, based on a fraction of the data. Periodically, after some number of local optimization steps, a node can initiate a pairwise interaction with a uniform random neighbor. Our main finding is that this procedure can converge even though the nodes can take several local steps between interactions, may perform *asynchronous* communication, reading stale versions of each others' models, and may compress data transmission through quantization. However, both in theory and practice, we observe trade-offs between convergence rate and degree of synchronization, in that the algorithm may need to perform additional gradient steps in order to attain a good solution, relative to the sequential baseline.

Our algorithm, called SWARMSGD, is decentralized in sense that each node maintains local version of the model, and two interacting nodes only see each others' models. We further allow that the data distribution at the nodes may not be i.i.d. Specifically, each node $i$ is assigned a set of samples $S^i$, and maintains its own parameter estimate $x^i$. Each node $i$ performs local SGD steps on its model $x^i$ based on its local data, and then picks a neighbor uniformly at random to share information with, by *averaging* of the two models. (To streamline the exposition, we ignore quantization and model staleness unless otherwise specified.) Effectively, if node $i$ interacts with node $j$, node $i$'s updated model becomes

$$x_{t+1}^i \leftarrow \frac{x_{t,H_i}^i + x_{t,H_j}^j}{2}, \tag{1}$$

where $t$ is the *total* number of interactions performed by all nodes up to this point, $j$ is the interaction partner of $i$ at step $t+1$, and the input models $x_{t,H_i}^i$ and $x_{t,H_j}^j$ have been obtained by iterating the SGD step $H_i$ and $H_j$ times, respectively, locally from the previous interaction of either node. We assume that $H_i$ and $H_j$ are random variables with mean $H$, that is, each node performs $H$ local steps in expectation between two communication steps. The update for node $j$ is symmetric, so that the two models match after the averaging step. In this paper, we analyze variants of the above SwarmSGD protocol.

The main intuition behind the algorithm is that the independent SGD steps will allow nodes to explore local improvements to the objective function on their subset of the data, while the averaging steps provide a decentralized way for the models to converge jointly, albeit in a loosely coupled way. We show that, as long as the maximum number of local steps is bounded, this procedure still converges, in the sense that gradients calculated at the average over all models are vanishing as we increase the number of interactions.

Specifically, assuming that the $n$ nodes each take a constant number of local SGD steps on average before communicating, we show that SwarmSGD has $\Theta(\sqrt{n})$ speedup to convergence in the non-convex case. This matches results from previous work which considered decentralized dynamics but which synchronized upon every SGD step, e.g. (Lian et al., 2017; 2018). Our analysis also extends to arbitrary regular graph topologies, non-blocking (delayed) averaging of iterates, and quantization. Generally, we show that the impact of decentralization, asynchrony, quantization, and local updates can be asymptotically negligible in reasonable parameter regimes.

On the practical side, we show that this algorithm can be mapped to a distributed system setting, where agents correspond to compute nodes, connected by a dense communication topology. Specifically, we apply SwarmSGD to train deep neural networks on image classification and machine translation (NMT) tasks, deployed on the Piz Daint supercomputer (Piz, 2019). Experiments confirm the intuition that the average synchronization cost of SwarmSGD per iteration is low: it stays around 10% or less of the batch computation time, and remains constant as we increase the number

of nodes. For example, using SwarmSGD deployed on 16 nodes, we are able to train a Transformer-XL (Vaswani et al., 2017) model on WMT17 (En-Ge) $1.5\times$ faster than a highly-optimized large-batch SGD baseline, and to *slightly higher accuracy*, without additional hyper-parameter tuning. At the same time, our method appears to be faster and more accurate than the previous practical decentralized methods, e.g. (Lian et al., 2017; 2018; Assran et al., 2018), in the same setting.

Importantly, we also note a negative result: in less overparametrized settings such as training residual CNNs (He et al., 2016) on ImageNet (Russakovsky et al., 2015), nodes do need to perform more iterations over the dataset relative to the baseline in order to recover full accuracy. This is predicted by the analysis, and confirms similar findings in previous work (Assran et al., 2018). Overall, however, our family of methods should be well-suited to training very large modern models in large-scale settings, where global synchronization among all nodes is prohibitively expensive.

**Related Work.** The study of decentralized optimization algorithms dates back to Tsitsiklis (1984), and is related to the study of *gossip* algorithms for information dissemination (Kempe et al., 2003; Xiao and Boyd, 2004; Boyd et al., 2006). Gossip is usually studied in one of two models (Boyd et al., 2006): *synchronous*, structured in global rounds, where each node interacts with a randomly chosen neighbor, and *asynchronous*, where each node wakes up at times given by a local Poisson clock, and picks a random neighbor to interact with. The model we consider can be seen as equivalent to the asynchronous gossip model. The key differences between our work and averaging in the gossip model, e.g. Boyd et al. (2006), are that that 1) we consider local SGD steps, which would not make sense in the case of averaging fixed initial values; and 2) the gossip input model is *static* (node inputs are fixed, and node estimates must converge to the true mean), whereas we study a *dynamic* setting, where models are continually updated via SGD. Several optimization algorithms have been analyzed in this setting (Nedic and Ozdaglar, 2009; Johansson et al., 2009; Shamir and Srebro, 2014), while Tang et al. (2018); Koloskova et al. (2019) analyze quantization in the *synchronous* gossip model.

Lian et al. (2017; 2018) and Assran et al. (2018) considered SGD-type algorithms in gossip-like models. Specifically, they analyze the SGD averaging dynamic in the non-convex setting *but do not allow nodes to perform local updates or quantize*. In particular, nodes perform pairwise averaging upon every SGD step. Table 2 in the Appendix provides a thorough comparison of assumptions, results, and rates. Their results are phrased in the synchronous gossip model, in which nodes interact in a sequence of perfect matchings, for which they provide $O(1/\sqrt{Tn})$ convergence rates under analytical assumptions. Lian et al. (2018) extends these results to a variant of the gossip model where updates can be performed based on stale information, similarly to our non-blocking extension.

Upon careful examination, one can find that their results can be extended to the asynchronous gossip setting we consider, *as long as nodes are not allowed to perform local SGD updates to their models (corresponding to $H = 1$) or to quantize communication*. Extending the analysis of distributed SGD to allow for local steps is challenging even in *centralized* models, see for instance Stich (2018). If we assume $H = 1$, our technique yields similar or better bounds relative to previous work in the decentralized model, as our potential analysis is specifically-tailored to this dynamic interaction model. For instance, for Assran et al. (2018), the speedup with respect to the number of nodes depends on a parameter $C$, which in turn, depends on 1) the dimension $d$ of the objective function, 2) the number of iterations for the graph given by edge sets of all matrices used in averaging to be connected, and the 3) diameter of the aforementioned connected graph. In the dynamic interaction model we consider, the parameter $C$ will be at least *linear* in the number of nodes $n$, which will eliminate any speedup. We present a systematic comparison in Appendix B.

In sum, relative to prior work on decentralized algorithms, our contributions are as follows. We are the first to consider the impact of local updates, asynchrony, and quantization in conjunction with decentralized SGD. We show that the cost for the linear reduction in communication in $H$ given by local steps is at worst a squared variance increase in the parameter $H$. Our analysis technique relies on a fine-grained analysis of individual interactions, which is different than that of previous work, and can yield improved bounds even in the case where $H = 1$. By leveraging the lattice-based quantization scheme of Davies et al. (2020), we also allow for communication-compression. From the implementation perspective, the performance of our algorithm is superior to that of previous methods, notably D-PSGD (Lian et al., 2017), AD-PSGD (Lian et al., 2018) and SGP (Assran et al., 2018), mainly due to the ability to take local steps.

Wang and Joshi (2018) and Koloskova et al. (2020) provide analysis frameworks for the synchronous version of decentralized SGD with local updates, and possibly changing topologies. This is a different setting from ours, since it requires each agent to take an equal number of gradient

steps before every interaction round, and therefore does not allow for agents to progress at different speeds (asynchrony). Further, we support quantization, and validate our analysis at scale.

## 2 PRELIMINARIES

**The Distributed System Model.** We consider a model which consists of $n \geq 2$ anonymous agents, or nodes, each of which is able to perform local computation. We assume that communication network of nodes is a $r$-regular graph $G$ with spectral gap $\lambda_2$, which denotes the second smallest eigenvalue of the Laplacian of $G$. This choice of communication topology models supercomputing and cloud networks, which tend to be regular, densely connected and low-diameter, mimicking regular expanders (Kim et al., 2008; Besta and Hoefler, 2014).

The execution proceeds in discrete *steps*, where in each step we sample an edge of the graph $G$ uniformly at random and we allow the agents corresponding to the edge endpoints interact. Each of the two chosen agents updates its state according to a state update function, specified by the algorithm. The basic unit of time is a single pairwise interaction between two nodes. Notice however that in a real system $\Theta(n)$ of these interactions could occur in parallel. Thus, a standard global measure is *parallel time*, defined as the total number of interactions divided by $n$, the number of nodes. Parallel time intuitively corresponds to the *average* number of interactions per node to convergence. We note that our model is virtually identical to the population model of distributed computing (Angluin et al., 2006), or to asynchronous gossip models (Xiao and Boyd, 2004).

**Stochastic Optimization.** We assume that the agents wish to minimize a $d$-dimensional, differentiable function $f : \mathbb{R}^d \to \mathbb{R}$. Specifically, we will assume the empirical risk minimization setting, in which agents are given access to a set of $m$ data samples $S = \{s_1, \ldots, s_m\}$ coming from some underlying distribution $\mathcal{D}$, and to functions $\ell_i : \mathbb{R}^d \to \mathbb{R}$ which encode the loss of the argument at the sample $s_i$. The goal of the agents is to converge on a model $x^*$ which minimizes the empirical loss over the $m$ samples, that is

$$x^* = \operatorname{argmin}_x f(x) = \operatorname{argmin}_x (1/m) \sum_{i=1}^{m} \ell_i(x). \tag{2}$$

In this paper, we assume that the agents employ these samples to run a decentralized variant of SGD, described in detail in the next section. For this, we will assume that each agent $i$ has access to *stochastic gradients* $\widetilde{g}_i$ of the function $f$, which are functions such that

$$\mathbb{E}[\widetilde{g}_i(x)] = \nabla f(x). \tag{3}$$

Stochastic gradients can be computed by each agent by sampling i.i.d. the distribution $\mathcal{D}$, and computing the gradient of $f$ at $\theta$ with respect to that sample. (Our analysis can be extended to the case where each agent is sampling from its own partition of data, see Section H in the Appendix.) We will assume a the following conditions about the objective function (One of the extensions removes the second moment bound):

1. **Smooth Gradients**: The gradient $\nabla f(x)$ is $L$-Lipschitz continuous for some $L > 0$, i.e. for all $x, y \in \mathbb{R}^d$:
$$\|\nabla f(x) - \nabla f(y)\| \leq L\|x - y\|. \tag{4}$$

2. **Bounded Second Moment**: The second moment of the stochastic gradients is bounded by some $M^2 > 0$, i.e. for all $x \in \mathbb{R}^d$ and agent $i$:
$$\mathbb{E}\left\|\widetilde{g}_i(x)\right\|^2 \leq M^2. \tag{5}$$

Note that throughout this paper for any random variable $X$, by $\mathbb{E}\|X\|^2$ we mean $\mathbb{E}[\|X\|^2]$.

## 3 THE SWARMSGD ALGORITHM

**Algorithm Description.** We now describe a decentralized variant of SGD, designed to be executed by a population of $n$ nodes, interacting over the edges of $r$-regular graph $G$. We assume that each node $i$ has access to local stochastic gradients $\widetilde{g}^i$, and maintains a model estimate $X^i$. For simplicity, we will assume that this initial model is $0^d$ at each agent, although its value may be arbitrary. Each

agent performs SGD steps on its local estimate $X^i$. At random times given by a clock of Poisson rate, we pick two neighboring agents $i$ and $j$ uniformly at random from $G$, and have them average their estimates. The interaction is precisely described in Algorithm 1.

For simplicity, the pseudocode is sequential, although in practice nodes perform their local SGD steps in parallel. Also, we have assumed a constant learning rate; we will detail the update procedure in the next section, as well as more complex variants of this basic update.

---

**Algorithm 1** Sequential SwarmSGD pseudocode for each interaction between nodes $i$ and $j$.

---
 % Let $G$ be $r$-regular graph.
 % Sample an edge $(i, j)$ of $G$ uniformly at random.
**Require:** agents $i$ and $j$ chosen for interaction
 % choose $H_i$ and $H_j$
 % agent $i$ performs $H_i$ local SGD steps
 **for** $q = 1$ **to** $H_i$ **do**
  $X^i \leftarrow X^i - \eta \widetilde{g}^i(X^i)$
 **end for**
 % agent $j$ performs $H_j$ local SGD steps
 **for** $q = 1$ **to** $H_j$ **do**
  $X^j \leftarrow X^j - \eta \widetilde{g}^j(X^j)$
 **end for**
 % agents *average* their estimates coordinate-wise
 $avg \leftarrow (X^i + X^j)/2$
 $X^i \leftarrow X^j \leftarrow avg$

---

## 4 THE CONVERGENCE OF SWARMSGD

We begin by analyzing the convergence of the baseline SwarmSGD algorithm. Fix an integer $H \geq 1$. First, we will consider a variant where $H_i$ and $H_j$ are independent, geometrically-distributed random variables, with mean $H$. This corresponds to interaction times being chosen by a Poisson clock of constant rate. To handle the fact that the number of local steps upon an interaction is a random variable, in this first case we will require stochastic gradients to satisfy the *bounded second moment assumption*, specified above. Intuitively, this is required since otherwise the "distance travelled" by a node could be virtually unbounded. In this setting, we prove the following:

**Theorem 4.1.** *Let $f$ be an non-convex, $L$-smooth function, whose stochastic gradients satisfy the bounded second moment assumption above. Let the number of local stochastic gradient steps performed by each agent upon interaction be a geometric random variable with mean $H$. Let the learning rate we use be $\eta = n/\sqrt{T}$. Define $\mu_t = \sum_{i=1}^{n} X_t^i/n$, where $X_t^i$ is a value of model $i$ after $t$ interactions, be the average of the local parameters. Then, for learning rate $\eta = n/\sqrt{T}$ and any number of interactions $T \geq n^4$:*

$$\frac{1}{T} \sum_{t=0}^{T-1} \mathbb{E}\|\nabla f(\mu_t)\|^2 \leq \frac{4(f(\mu_0) - f(x^*))}{\sqrt{T}H} + \frac{2304H^2 \max(1, L^2)M^2}{\sqrt{T}} \left( \frac{r^2}{\lambda_2^2} + 1 \right).$$

**Discussion.** First, we note that this notion of convergence is standard in the non-convex case, e.g. (Lian et al., 2015; 2017; 2018), and that each of the upper bound terms has an intuitive interpretation: the first represents the reduction in loss relative to the initialization, and gets *divided* by the number of local steps $H$, since progress is made in this term in every local step; the second represents the influence of the variance of each individual local step multiplied by a term which bounds the impact of the graph topology on the convergence. In particular, this term negatively impacts convergence for large values of $H$, $L$, and $M$, but gets dampened if the graph is well-connected (i.e. large $\lambda_2$). For example, in the case of the complete graph, we have $\lambda_2 = n$.

Second, let us consider the algorithm's communication complexity, which we measure in terms of the total number of communication steps. We notice an interesting trade-off between the linear reduction in $H$ in the first term of the bound, showing that the algorithm takes advantage of the local gradient steps, and the quadratic increase in the second variance term also due to $H$, in the second term. Hence, the end-to-end speedup of our algorithm versus the variant with $H = 1$ will depend on the relationship between these two terms, which depends on the parameter values.

Third, importantly, the time $T$ in this bound counts the *total* number of interactions. However, in practice $\Theta(n)$ pairwise interactions will occur in parallel, as they are independent. Therefore, we can replace $T$ by $nT$ in the above formula, to estimate the speedup in terms of wall-clock time, obtaining a speedup of $\Theta(\sqrt{n})$. At the same time, notice that this speedup is dampened in the second term by the non-trivial additional variance due to noisy local gradient steps, a fact which we will revisit in the experimental section.

Fourth, although the requirement $T \geq n^4$ appears restrictive, some non-trivial dependency between $n$ and $T$ is necessary, as gradient information has to "mix" well in the graph before global optimization can occur. Previous work requires stronger variants of this restriction: specifically, Lian et al. (2018) require $T \geq n^6$, while Assran et al. (2018) requires $T = \Omega(nd^2)$.

**Proof Overview.** At a high level, the argument rests on two technical ideas. The first idea is to show that, due to the pairwise averaging process, and in spite of the local steps, the nodes' parameters will have to remain concentrated around their mean $\mu_t$. The second is to show that, even though stochastic gradients are taken at *perturbed, noisy* estimates of this mean, the impact of this noise on convergence can be bounded.

In particular, the main technical difficulty in the proof is to correctly "encode" the fact that parameters are well concentrated around the mean. For this, we define the potential $\Gamma_t$, which denotes the variance of models after $t$ interactions. Formally,

$$\Gamma_t = \sum_{i=1}^{n} \|X_t^i - \mu_t\|^2, \tag{6}$$

where $\mu_t = \sum_{i=1}^{n} X_t^i/n$. We bound the expected evolution of $\Gamma_t$ in terms of $r$, the degree of nodes in the interaction graph $G$, and $\lambda_2$, the second smallest eigenvalue of the Laplacian of $G$. For both algorithm variants we consider, our bound depends on the learning rate, number of local steps, and the bound provided by the assumption on the stochastic gradients (the bound $M^2$). The critical point is that the upper bound on the expectation of $\Gamma_t$ *does not depend* on the number of interactions $t$. Our approach leverages techniques from the analysis of static load balancing schemes, e.g. Berenbrink et al. (2009). Two key elements of novelty in our case are that (1) for us the load balancing process is *dynamic*, in the sense that loads (gradients) get continually added; (2) the load-balancing process we consider is multi-dimensional, whereas usually the literature considers simple scalar weights. The complete argument is presented in the Appendix.

This technique is quite powerful, as it allows for a number of non-trivial extensions:

**Extension 1: Removing the second-moment bound and allowing for non-i.i.d. local data.** In the first extension of the algorithm, we assume that the number of local steps performed by each agent is fixed and is equal to $H$. In this case, we are able to remove the bounded second moment assumption, and are able to prove convergence under standard assumptions for non-i.i.d data. Specifically, in the non-i.i.d. setting, we consider that each $f_i(x)$ is the local function of agent $i$ (computed over the samples available to $i$). We will require that 1) the function $f_i$ is $L$-smooth, and that 2) for each agent $i$, $\tilde{g}_i$ is unbiased estimate of $f_i$ and that 3) for any $x$, $\mathbb{E}[\|\tilde{g}_i(x) - f_i(x)\|^2] \leq \sigma^2$. We define $f(x) = \sum_{i=1}^{n} f_i(x)/n$ and the bound $\sum_{i=1}^{n} \|\nabla f_i(x) - \nabla f(x)\|^2/n \leq \rho^2$.

**Theorem 4.2.** *Let $f$ be an non-convex, $L$-smooth function whose minimum $x^\star$ we are trying to find via the SwarmSGD procedure given in Algorithm 1. Assume the local functions of agents satisfy the conditions discussed above. Let $H$ be the number of local stochastic gradient steps performed by each agent before interacting. Define $\mu_t = \sum_{i=1}^{n} X_t^i/n$, where $X_t^i$ is a value of model $i$ after $t$ interactions. For learning rate $\eta = \frac{n}{\sqrt{T}}$ and $T = \Omega\left(n^4 H^2 \max(1, L^2)\left(\frac{r^2}{\lambda_2^2} + 1\right)^2\right)$ we have that:*

$$\frac{\sum_{i=0}^{T-1} \mathbb{E}\|\nabla f(\mu_t)\|^2}{T} \leq \frac{1}{\sqrt{T}H}\mathbb{E}[f(\mu_0) - f(x^\star)] + \frac{376 H^2 \max(1, L^2)(\sigma^2 + 4\rho^2)}{\sqrt{T}}\left(\frac{r^2}{\lambda_2^2} + 1\right).$$

Please see Appendix H for the details of the proof; we note that we did not optimize for constants. Relative to Theorem 4.1, we have the same quadratic dependency on the number of local steps $H$ and on $L$, but now the second moment bound is replaced by the variance terms. We emphasize that for non-i.i.d data under second-moment bounds, the exact same bounds as in Theorem 4.1 will hold.

**Extension 2: Non-blocking averaging.** Algorithm 1 is *blocking*, in that it requires both nodes to complete their local iterations at the same time before they can interact. In practice, nodes can

average their local updates without synchronizing, as follows. Each node $i$ keeps two copies of the model: the *live copy* $X_i$ (on which local SGD iterations are applied) and the *communication copy* $Y_i$, which can be accessed asynchronously by communicating nodes. When completing its local steps, a node $i$ first checks if some other node averaged against its *communication copy* $Y_i$ since its last communication step. If the answer is yes, it simply applies its locally-generated update to its communication model $Y_i$, updates the live copy so that $X_i = Y_i$, and proceeds with the next iteration of local computation. If no other node has averaged against its communication copy, then the node actively seeks a random communication partner $j$, and averages its live copy against its model $Y_j$, updating both to $(X_i + Y_j)/2$. The node then proceeds with the next iteration of local computation. Please see Appendix F for the precise definition of the algorithm, and for the formal convergence guarantee for this variant.

**Extension 3: Quantization.** For large models, the cost of the averaging step can become significant, due to bandwidth constraints. To remove the bandwidth bottleneck, we allow the averaging step to be performed with respect to *quantized* versions of the two models. While communication-compression has been considered in a decentralized context before, e.g. (Tang et al., 2018; Lu and Sa, 2020), our approach is different. Instead of modifying the algorithm or maintaining neighbor information at every node, we make use of a quantization scheme with some useful properties by Davies et al. (2020), which we slightly adapt to our context.

The key issue when quantizing decentralized models is that for most known quantization schemes, e.g. (Alistarh et al., 2017), the quantization error depends on the *norm of the inputs*: here, the inputs are models, which are not necessarily close to the origin. Thus, the quantization error at each step would depend on the norm of the models, which would break our bound on $\Gamma_t$. Instead, we observe that the quantization scheme of Davies et al. (2020) has error which is bounded by the *distance between inputs*, rather than input norms. Crucially, we show that $\Gamma_t$ can in fact be used to bound the distance between models, so we can bound the quantization error in terms of $\Gamma_t$ at each step. This allows us, with some care, to generalize the analysis to the case where the models are quantized. We provide a full description and proof of convergence in Appendix G.

Specifically, quantization ensures the same convergence bounds as in Theorem 4.1, but with an expected communication cost of $O(d + \log T)$ bits per step.[1] By contrast, non-quantized decentralized algorithms assume that nodes can exchange infinite-precision real numbers, while the only other memory-less compression scheme (Lu and Sa, 2020) induces a linear dependence in $d$ in the rate. In our applications, $d \gg \log T$, and therefore our cost is essentially constant per dimension; specifically, we show that we can quantize to 8 bits per coordinate without loss of accuracy.

## 5 EXPERIMENTAL RESULTS

In this section, we validate our analysis, by applying the algorithm to training deep neural networks for image classification and machine translation. We map the algorithm onto a multi-node supercomputing setting, in which we have a large number of compute nodes, connected by fast communication links. The key overhead in this setting is *synchronization*: at large node counts, the cost of synchronizing all nodes so they execute in lock-step can be very high, see e.g. Li et al. (2019) for numerical results on different workloads. SwarmSGD mitigates this overhead, since nodes synchronize only sporadically and in pairs. Harnessing the computational power of this large-scale distributed setting is still an underexplored area (Ben-Nun and Hoefler, 2019).

**Target System and Implementation.** We run SwarmSGD on the CSCS Piz Daint supercomputer, which is composed of Cray XC50 nodes, each with a Xeon E5-2690v3 CPU and an NVIDIA Tesla P100 GPU, using a state-of-the-art Aries interconnect over a Dragonfly network topology, which is *regular*. Please see (Piz, 2019) for more details. We implemented SwarmSGD in Pytorch and TensorFlow using NCCL and MPI-based primitives. Both variants implement the version with non-blocking averaging. The Pytorch implementation is on top of SGP framework (Assran et al., 2018), and uses SwarmSGD to train ResNets on the CIFAR-10/100 (Krizhevsky et al., 2014) and ImageNet (Russakovsky et al., 2015) datasets, while we use the TensorFlow implementation to train a much larger Transformer-XL model (Vaswani et al., 2017) on the WMT17 (En-Ge) dataset. We note that all algorithms used the same topology overlay (fully-connected with random pairings), and that SGP was run with overlap factor 1, as suggested by Assran et al. (2018).

---

[1] The unusual $\log T$ factor arises because the quantization scheme of Davies et al. (2020) can *fail* with some probability, which we handle as part of the analysis.

| Model / Dataset | SGD Top-1 | LB SGD Top-1 | SwarmSGD | Parameters |
|---|---|---|---|---|
| ResNet20 / CIFAR-10 | 91.7% (200 epochs) | 91.5% (200 epochs) | 91.79% (280 epochs) | 4 local steps |
| ResNet18 / ImageNet | 69.76 % (90 epochs) | 69.17% (90 epochs) | 69.79% (240 epochs) | 3 local steps |
| ResNet50 / ImageNet | 76.14% (90 epochs) | 75.43% (90 epochs) | 75.68% (240 epochs) | 2 local steps |

Table 1: Parameters for full Top-1 validation accuracy on CIFAR-10 and ImageNet running on 32 nodes. Swarm step count represents *local SGD steps per model* between two averaging steps, and epochs are counted in terms of total passes over the data.

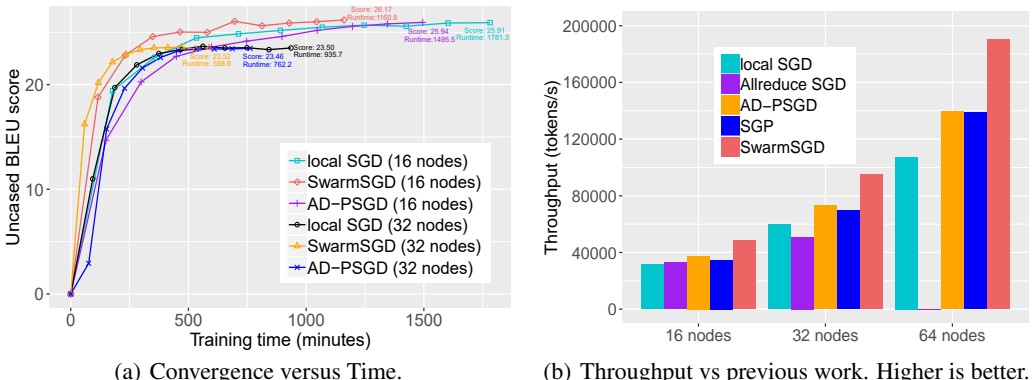

(a) Convergence versus Time.     (b) Throughput vs previous work. Higher is better.

Figure 1: Convergence and Scalability on the Transformer/WMT Task with multiplier = 1.

**Training Process.** Our training methodology follows data-parallel training, with some differences due to decentralization, and is identical to previous work on decentralized and local SGD, e.g. (Lian et al., 2017; Assran et al., 2018; Lin et al., 2018). Training proceeds in *epochs*, each of which corresponds to processes collectively performing a full pass over the dataset. At the beginning of each epoch, we re-shuffle the dataset and partition it among processes (Lin et al., 2018).

As noted in previous work (Lian et al., 2017; 2018; Assran et al., 2018) variants of decentralized SGD are not always able to recover sequential SGD accuracy within the same number of epochs as this baseline. This is justified by Theorems 4.1 and 4.2, which predict that the slower mixing (and higher local model variance) can affect convergence. Thus, in some experiments, we will allow the decentralized schemes to execute for more epochs, by a constant *multiplier* factor between 1 and 3. Once we have fixed the number of epochs, *we do not alter the other training hyperparameters*: in particular, the learning rate schedule, momentum and weight decay terms are identical to sequential SGD, for each individual model.

**Accuracy and Speed.** We first examined whether SwarmSGD can in fact recover full accuracy versus the sequential or large-batch SGD baselines. In Table 1 we provide an overview of parameter values to recover or exceed large-batch SGD accuracy (following (Goyal et al., 2017)) using SwarmSGD, on the ResNet/ImageNet/CIFAR tasks. We execute for 32 nodes on ImageNet, and 8 nodes on CIFAR-10. (Local batch sizes are 256 for ResNet20 and ResNet18, and 128 for ResNet50. Quantization is not applied.) The results show that Swarm can recover or slightly exceed the accuracy of the large-batch baselines, and that it has lower practical communication cost relative to existing methods (see Figure 2(b), where we separate the average computation cost per batch). However, Swarm requires significant additional passes over the data (up to $2.7\times$) to achieve full accuracy, which negates its performance benefits in this specific setting, relative to large-batch SGD. (Please see Appendix Figure 5 for an end-to-end time comparison. We do not take the cost of fine-tuning the hyperparameters for large-batch SGD into account in this example.) This finding is in line with previous work on decentralized methods (Assran et al., 2018).

Next, we examine accuracy for the WMT17 task. The results are provided in Figure 1(a), in accuracy-vs-time format, for 16 and 32 nodes, executing for 10 global epochs. Here, the large-batch SGD (LB-SGD) baseline (BLEU score 26.1 at 16 nodes) is a poor alternative at high node counts: its throughput is very low, due to the size of the model (see Figure 1(b)). At 16 nodes, Swarm

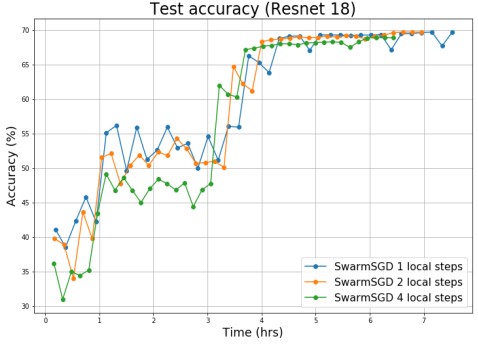
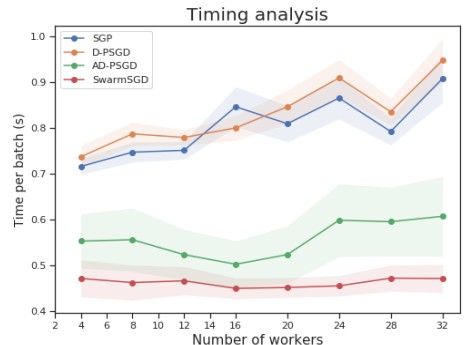

(a) Convergence in time versus number of local steps for ResNet18 on ImageNet. All variants recover the target accuracy, but we note the lower convergence of variants with more local steps. The experiment is run on 32 nodes.

(b) Average time per batch for previous methods, compared to SwarmSGD, on ResNet18/ImageNet. The base value on the y axis (0.4) is the average computation time per batch, so values above represent average communication time per batch.

Figure 2: Convergence results and performance breakdown for ResNet18/ImageNet.

*slightly exceeds* the baseline accuracy at 26.17 BLEU, for an end-to-end speedup of $\sim 1.5\times$. In the same setting, Swarm outperforms all other decentralized methods (the fastest previous method, AD-PSGD, is 30% slower, and less accurate), both in terms of BLEU score, and in terms of end-to-end time. (The objective loss graph is similar, and is given in Appendix Figure 7.) At 32 nodes, all decentralized methods reach lower scores ($\sim 23.5$) after 10 epochs. However, we observed experimentally that running Swarm for an additional 5 epochs at 32 nodes recovered a BLEU score of $\sim 25.9$, 30% faster than the 16-node version in terms of end-to-end time (omitted for visibility).

In addition, we investigated 1) the accuracy of the *real average* of all models throughout training: it is usually more accurate than an arbitrary model, but not significantly, corroborating the claim that individual models tend to stay close to the mean; 2) the influence of the number of local steps on accuracy: perhaps surprisingly, we were able to recover baseline accuracy on ResNet18/ImageNet for up to 4 local steps (see Figure 2(a)); 3) the impact of quantization on convergence, where we were able to recover accuracy when applying 8-bit model quantization to Swarm. We encourage the reader to examine the full experimental report in the Appendix, which contains data on these experiments, as well as additional ablation studies.

**Discussion.** Generally, the performance and accuracy of SwarmSGD are superior to previous decentralized methods (see Figure 1 for an illustration, and Figure 2(b) for a performance breakdown). In particular, a closer examination of the average batch times in Figure 2(b) shows that time per node per batch (including communication and computation) is largely *constant* as we increase the number of nodes, which gives our method close-to-ideal scaling behaviour. This advantage relative to previous schemes, notably AD-PSGD, comes mainly from the *reduction in communication frequency*: Swarm communicates less often, and therefore incurs lower average communication cost.

The main disadvantage of Swarm is that, similar to previous decentralized methods, it may need additional data passes in order to fully recover accuracy at high node counts. However, we also note that our method did not benefit from the high level of hyperparameter tuning applied to large-batch SGD, e.g. (Goyal et al., 2017). We find it interesting that this accuracy issue is less prevalent in the context of large, over-parameterized models, such as the Transformer, where Swarm could be a practically-viable alternative to large-batch SGD within the same number of epochs.

## 6    CONCLUSIONS AND FUTURE WORK

We analyzed the convergence of SGD in a decoupled model of distributed computing, in which nodes mostly perform independent SGD updates, interspersed with intermittent pairwise averaging steps, which may be performed in an inconsistent and noisy manner. We showed that SGD still converges in this restrictive setting, and under considerable consistency relaxations, and moreover can still achieve speedup in terms of iteration time. Empirical results in a supercomputing environment complement and validate our analysis, showing that this method can outperform previous proposals. A natural extension would be to generalize the bounds to arbitrary communication graphs, or in terms of the assumptions on the objective, or to experiment on large-scale decentralized testbeds.

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

## A  SUMMARY OF THE APPENDIX SECTIONS

Appendix contains the following sections:

- In **Section B** we compare SwarmSGD with some of the existing algorithms. We list convergence bounds and the assumptions needed to achieve them.

- In **Section C** we provide crucial properties for the load balancing on the graph.

- In **Section D** we provide definitions for the local steps we use in the later sections.

- In **Section E** we provide the sketch of proof of Theorem 4.1, which shows the convergence of SwarmSGD assuming the second moment bound on the gradients. Recall that the number of local steps in this case is a geometric random variable with mean $H$.

- In **Section F** we provide the proof for the non-blocking version of the swarm SGD algorithm. We again assume the second moment bound on the gradients and that the number of local steps is a geometric random variable with mean $H$.

- In **Section G** we provide the proof for the quantized version of the swarm SGD algorithm. We again assume the second moment bound on the gradients and that the number of local steps is a geometric random variable with mean $H$.

- In **Section H** we prove Theorem 4.2. In this case we do not assume the second moment bound, data is not distributed identically and the number of the local steps performed by each agent is a fixed number $H$.

- In **Section I** we provide additional experimental results for SwarmSGD.

## B  COMPARISON OF RESULTS

In this section we compare convergence rates of existing algorithms, while specifying the bounds they require for convergence. In the tables $T$-corresponds to the parallel time and $n$ is a number of processes. We use the following notations for needed bounds (or assumptions):

1. $\sigma^2$ - bound on the variance of gradient .

2. $M^2$ - bound on the second moment of gradient.

3. $d$ - bounded dimension.

4. $\lambda_2$ - bounded spectral gap of the averaging matrix (interaction graph in case of SwarmSGD).

5. $\tau$ - bounded message delay.

6. $r$ - interaction graph is $r$-regular.

7. $\Delta$ - bounded diameter of interaction graph.

| Algorithm | Assumptions | Convergence Rate |
|---|---|---|
| SwarmSGD | $\sigma^2, \lambda_2, r$ | $O(1/\sqrt{Tn})$ |
| SwarmSGD | $M^2, \lambda_2, r$ | $O(1/\sqrt{Tn})$ |
| AD-PSGD Lian et al. (2018) | $\sigma^2, \lambda_2, \tau$ | $O(1/\sqrt{Tn})$ |
| SGP Assran et al. (2018) | $\sigma^2, d, \Delta, \tau$ | $O(1/\sqrt{Tn})$ |

Table 2: **Comparison of theoretical results in the non-convex case.**

**Discussion.** We compare in more detail against Lian et al. (2018) and Assran et al. (2018), since these are the only other papers which do not require explicit global synchronization in the form of rounds. (By contrast, e.g. Wang and Joshi (2018); Koloskova et al. (2020) require that nodes synchronize in rounds, so that at every point in time each node has taken the same number of steps.)

In Assran et al. (2018), all nodes perform gradient steps at each iteration, but averaging steps can be delayed by $\tau$ iterations. Unfortunately, in this case the mixing time depends on the dimension - $d$

(more precisely, it contains a $\sqrt{d}$ factor!), on the delay bound $\tau$, and on $\Delta$, defined as the number of iterations over which the interaction graph is well connected. Additionally, the analysis is not suitable for random interactions. On the other hand, Lian et al. (2018) consider random interaction matrices and do not require the agents to perform the same number of gradient steps. Unlike our model, in their case more than two nodes can interact during the averaging step.

To circumvent the global synchronization issue, Lian et al. (2018) allow agents to have outdated views during the averaging step. Yet, we would like to emphasize that they require blocking during the averaging steps, while we allow some amount of non-blocking property. By some amount we means that algorithm needs blocking only in the case when some node takes more than two consecutive iterations to complete it's local gradient steps. This means that for each node $i$ to complete $H_i$ local steps should not more take more then $O(n)$ global steps (since each node interacts with probability $2/n$ at each step), this assumption also holds for Lian et al. (2018).

In summary, our algorithm reduces the synchronization required by averaging steps, by considering pairwise interactions and by introducing local steps and providing non-blocking version of the algorithm as well (in SGP and AD-PSGD, agents perform one local step and one averaging step per iteration). We would like to point out that we also allow a random number of local steps between interactions in the case when we have second moment bound on the stochastic gradient, which reduces synchronization costs even further. Finally, our algorithm requires $T \geq O(n^4)$ number of iterations to achieve the convergence rate of $O(1/\sqrt{Tn})$ in the case of blocking algorithm and $T = \Omega(n^6)$ in general. (By contrast, Lian et al. (2018) requires $T = \Omega(n^6)$.)

## C  PROPERTIES OF THE LOAD BALANCING

In this section provide the useful lemmas which will help as in the later sections.

We are given a simple undirected graph $G$, with $n$ nodes (for convenience we number them from 1 to $n$) and edge set $E$. Each node is adjacent to exactly $r$ nodes.

Each node $i$ of graph $G$ keeps a local vector model $X_t^i \in \mathbb{R}^d$ ($t$ is the number of interactions or steps); let $X_t = (X_t^1, X_t^2, ..., X_t^n)$ be the vector of local models at step $t$.

An interaction (step) is defined as follows: we pick an edge $e = (u, v)$ of $G$ uniformly at random and update the vector models correspondingly.

Let $\mu_t = \sum_{i=1}^n X_t^i/n$ be the average of models at step $t$ and let $\Gamma_t = \sum_{i=1}^n \|X_t^i - \mu_t\|^2$ be a potential at time step $t$.

Let $\mathcal{L}$ be the Laplacian matrix of $G$ and let let $\lambda_2$ be a second smallest eigenvalue of $\mathcal{L}$. For example, if $G$ is a complete graph $\lambda_2 = n$.

First we state the following lemma from Ghosh and Muthukrishnan (1996):

**Lemma C.1.**

$$\lambda_2 = \min_{v=(v_1,v_2,...,v_n)} \left\{ \frac{v^T \mathcal{L} v}{v^T v} \Big| \sum_{i=1}^n v_i = 0 \right\} . \tag{7}$$

Now, we show that Lemma C.1 can be used to lower bound $\sum_{(i,j)\in E} \|X_t^i - X_t^j\|^2$:

**Lemma C.2.**

$$\sum_{(i,j)\in E} \|X_t^i - X_t^j\|^2 \geq \lambda_2 \sum_{i=1}^n \|X_t^i - \mu_t\|^2 = \lambda_2 \Gamma_t. \tag{8}$$

*Proof.* Observe that

$$\sum_{(i,j)\in E} \|X_t^i - X_t^j\|^2 = \sum_{(i,j)\in E} \|(X_t^i - \mu_t) - (X_t^j - \mu_t)\|^2. \tag{9}$$

Also, notice that Lemma C.1 means that for every vector $v = (v_1, v_2, ..., v_n)$ such that $\sum_{i=1}^n v_i = 0$, we have:

$$\sum_{(i,j)\in E} (v_i - v_j)^2 \geq \lambda_2 \sum_{i=1}^n v_i^2. \tag{10}$$

Since $\sum_{i=1}^{n}(X_t^i - \mu_t)$ is a 0 vector, we can apply the above inequality to the each of $d$ components of the vectors $X_t^1 - \mu_t, X_t^2 - \mu_t, ..., X_t^n - \mu_t$ separately, and by elementary properties of 2-norm we prove the lemma. □

## D  DEFINITIONS FOR THE LOCAL STEPS

In this section we provide the formal definition of the local steps performed by our algorithms. Recall that $X_t^i$ is a local model of node $i$ at step $t$. Let $H_t^i$ be the number of local steps node $i$ performs in the case when it is chosen for interaction at step $t + 1$. A natural case is for $H_t^i$ to be fixed throughout the whole algorithm, that is: for each time step $t$ and node $i$, $H_t^i = H$. However, optimal choice $H_t^i$ depends on whether a second moment bound on gradients (5) is assumed. Let:

$$\widetilde{h}_i^0(X_t^i) = 0.$$

and for $1 \le q \le H_t^i$ let:

$$\widetilde{h}_i^q(X_t^i) = \widetilde{g}_i(X_t^i - \sum_{s=0}^{q-1} \eta\widetilde{h}_i^s(X_t^i)),$$

Note that stochastic gradient is recomputed at each step, but we omit the superscript for simplicity, that is: $\widetilde{h}_i^q(X_t^i) = \widetilde{g}_i^q(X_t^i - \sum_{s=0}^{q-1} \eta\widetilde{h}_i^s(X_t^i))$. Further , for $1 \le q \le H_t^i$, let

$$h_i^q(X_t^i) = \mathbb{E}[\widetilde{g}_i(X_t^i - \sum_{s=0}^{q-1} \eta\widetilde{h}_i^s(X_t^i))] = \nabla f(X_t^i - \sum_{s=0}^{q-1} \eta\widetilde{h}_i^s(X_t^i))$$

be the expected value of $\widetilde{h}_i^q(X_t^i)$ taken over the randomness of the stochastic gradient $\widetilde{g}_i$. Let $\widetilde{h}_i(X_t^i)$ be the sum of $H_t^i$ local stochastic gradients we computed:

$$\widetilde{h}_i(X_t^i) = \sum_{q=1}^{H_t^i} \widetilde{h}_i^q(X_t^i).$$

Similarly, for simplicity we avoid using index $t$ in the left side of the above definition, since it is clear that if the local steps are applied to model $X_t^i$ we compute them in the case when node $i$ interacts at step $t + 1$. The update step in Swarm SGD (Algorithm 1) is (before averaging):

$$X_{t+1}^i = X_t^i - \eta\widetilde{h}_i(X_t^i) = X_t^i - \eta\sum_{q=1}^{H_t^i} \widetilde{h}_i^q(X_t^i) = X_t^i - \eta\sum_{q=1}^{H_t^i} \widetilde{g}_i(X_t^i - \sum_{s=0}^{q-1} \eta\widetilde{h}_i^s(X_t^i)).$$

Notice that

$$\mathbb{E}\|\widetilde{h}_i^q(X_t^i)\|^2 = \mathbb{E}\|\widetilde{g}_i(X_t^i - \sum_{s=0}^{q-1} \eta\widetilde{h}_i^s(X_t^i))\|^2 \overset{\text{Assumption 5}}{\le} M^2. \tag{11}$$

## E  ANALYSIS UNDER SECOND MOMENT BOUND AND RANDOM NUMBER OF LOCAL STEPS

In this section we consider Algorithm 1, where for each node $i$, $H_i$ is a geometric random variable with mean $H$. We also assume a gradient second moment bound (5). We provide only a sketch of the proof since the proof for the non-blocking version of algorithm in Section F is more general. If nodes $i$ and $j$ interact at step $t + 1$ and their local models have values of $X_t^i$ and $X_t^j$ after step $t$. Their new model values become:

$$X_{t+1}^i = X_{t+1}^j = (X_t^i + X_t^j - \eta\widetilde{h}_i(X_t^i) - \eta\widetilde{h}_j(X_t^j))/2.$$

Recall that $\mu_t$ is average of the values of models after time step $t$ and $\Gamma_t = \sum_{i=1}^{n} \|X_t^i - \mu_t\|^2$ First of all we can prove that (see Lemma F.1, the result can be achieved even though algorithms differ):

$$\mathbb{E}[\Gamma_{t+1}] \le \mathbb{E}[\Gamma_t](1 - \frac{\lambda_2}{2rn}) + (2 + \frac{4r}{\lambda_2})\eta^2 \sum_{i=1}^{n} \left(4\mathbb{E}\|\widetilde{h}_i(X_t^i)\|^2\right). \tag{12}$$

We can further show that Lemma F.2, and therefore Lemmas F.3 and F.4, also hold, yielding:

$$\mathbb{E}[\Gamma_t] \leq (\frac{40r}{\lambda_2} + \frac{80r^2}{\lambda_2^2})n\eta^2 H^2 M^2, \tag{13}$$

and

$$\sum_{i=1}^{n} \mathbb{E}\langle \nabla f(\mu_t), -\widetilde{h}_i(X_t^i)\rangle \leq 2HL^2\mathbb{E}[\Gamma_t] - \frac{3Hn}{4}\mathbb{E}\|\nabla f(\mu_t)\|^2 + 12H^3nL^2M^2\eta^2. \tag{14}$$

Next in the similar fashion as in the proof of Theorem F.8 we can show that :

$$\mathbb{E}[f(\mu_{t+1})] \leq \mathbb{E}[f(\mu_t)] + \frac{2\eta}{n^2}\sum_{i=1}^{n} \mathbb{E}\langle \nabla f(\mu_t), -\widetilde{h}_i(X_t^i)\rangle + \frac{20L\eta^2 H^2 M^2}{n^2}. \tag{15}$$

This allows to show that:

**Theorem 4.1.** *Let $f$ be an non-convex, L-smooth function, whose stochastic gradients satisfy the bounded second moment assumption above. Let the number of local stochastic gradient steps performed by each agent upon interaction be a geometric random variable with mean $H$. Let the learning rate we use be $\eta = n/\sqrt{T}$. Define $\mu_t = \sum_{i=1}^{n} X_t^i/n$, where $X_t^i$ is a value of model $i$ after $t$ interactions, be the average of the local parameters. Then, for learning rate $\eta = n/\sqrt{T}$ and any number of interactions $T \geq n^4$:*

$$\frac{1}{T}\sum_{t=0}^{T-1} \mathbb{E}\|\nabla f(\mu_t)\|^2 \leq \frac{4(f(\mu_0) - f(x^*))}{\sqrt{T}H} + \frac{2304H^2\max(1, L^2)M^2}{\sqrt{T}}(\frac{r^2}{\lambda_2^2} + 1).$$

*Proof.* We again skip the calculations and follow steps from the proof of Theorem F (note that constants can be improved, but for simplicity we keep them the same). After applying (13) and (14), this results in:

$$\mathbb{E}[f(\mu_{t+1})] - \mathbb{E}[f(\mu_t)] \leq (\frac{160r}{\lambda_2} + \frac{320r^2}{\lambda_2^2})\frac{\eta^3 H^3 M^2 L^2 n}{n^2} - \frac{Hn}{4}\mathbb{E}\|\nabla f(\mu_t)\|^2$$
$$+ \frac{76H^3L^2M^2\eta^3}{n} + \frac{20L\eta^2 H^2 M^2}{n^2}.$$

once we sum up the above inequality $t = 0$ to $t = T - 1$ and massage terms we get (additionally recall that $\mathbb{E}[f(\mu_T)] \geq f(x^*)$):

$$\frac{1}{T}\sum_{t=0}^{T-1}\mathbb{E}\|\nabla f(\mu_t)\|^2 \leq \frac{4n(f(\mu_0) - f(x^*))}{TH\eta} + (\frac{640r}{\lambda_2} + \frac{1280r^2}{\lambda_2^2})\eta^2 H^2 M^2 L^2 T$$
$$+ \frac{80TL\eta HM^2}{n} + 304TH^2 L^2 M^2 \eta^2.$$

Finally since $\eta = n/\sqrt{T} \leq \frac{1}{n}$ (because $T \geq n^4$) we get the proof of the lemma. Note that the difference between this theorem and Theorem F.8 is that we have lower bound of $n^4$ for $T$ here, instead of $n^4(n + 1)^2$. The reason is that we are not required to use Lemmas F.7 and F.5 since our algorithm allows blocking and this means that interacting agents do not have incomplete values for the models. $\square$

## F  ANALYSIS OF THE NONBLOCKING VARIANT, WITH SECOND MOMENT BOUND AND RANDOM NUMBER OF LOCAL STEPS

First we define how the non-blocking property changes our interactions. Let $i$ and $j$ be nodes which interact at step $t + 1$, we set

$$X_{t+1/2}^i = \frac{X_t^i}{2} + \frac{X_t^{j'}}{2},$$
$$X_{t+1/2}^j = \frac{X_t^j}{2} + \frac{X_t^{i'}}{2},$$

and

$$X_{t+1}^i = X_{t+1/2}^i - \eta \widetilde{h}_i(X_t^i),$$
$$X_{t+1}^j = X_{t+1/2}^j - \eta \widetilde{h}_j(X_t^j),$$

where for each node $k$, if $p_t^k + 1$ is the last time interacting before and including step $t$:

$$X_t^{k'} = X_{p_t^k+1/2}^i = X_{p_t^k+1} + \eta \widetilde{h}_k(X_{p_t^k}^k) = X_t^k + \eta \widetilde{h}_k(X_{p_t^k}^k). \tag{16}$$

Intuitively the last definition means that node $k$ has computed $X_{p_t^k+1/2}^i$ but has not finished computing $X_{p_t^k+1}$, hence when some other node tries to read $X_{p_t^k+1}$, it reads the value which is missing local gradient update step, but it does not have to wait for node $k$ to finish computing. Since $p_t^k + 1$ is the last step node $k$ interacted we have that $X_t^k = X_{p_t^k+1}^k$. More formally:

---

**Algorithm 2** Sequential non-blocking SwarmSGD pseudocode for each interaction between nodes $i$ and $j$.

---

    % Let $G$ be $r$-regular graph.
    % Sample an edge $(i, j)$ of $G$ uniformly at random.
**Require:** agents $i$ and $j$ chosen for interaction, $i$ is initiator
    % choose $H_i$ and $H_j$
    % agent $i$ performs $H_i$ local SGD steps
    $S^i \leftarrow X_i$
    **for** $q = 1$ **to** $H_i$ **do**
        $X^i \leftarrow X^i - \eta \widetilde{g}^i(X^i)$
    **end for**
    % agent $j$ performs $H_j$ local SGD steps
    $S^j \leftarrow X^j$
    **for** $q = 1$ **to** $H_j$ **do**
        $X^j \leftarrow X^j - \eta \widetilde{g}^j(X^j)$
    **end for**
    % agents *update* their estimates
    $X^i \leftarrow (S^i + X^{j'})/2 + (X^i - S^i)$
    $X^j \leftarrow (S^j + X^{i'})/2 + (X^j - S^j)$

---

Notice the differences between the main algorithm and non-blocking one: first, local gradient steps are applied only after the averaging steps (this corresponds to term $X^i - S^i$ for node $i$), and second, nodes get access to the model of their interacting partner, which might not be complete for the reasons described above (for example, node $i$ is forced to use $X^{j'}$ instead of $X^j$ in its averaging step). If node $i$ is the initiator of the interaction and its chosen interaction partner $j$ is still computing the local gradients from its previous interaction, this algorithm allows node $i$ not to wait for $j$ to finish computation. In this case, $i$ simply leaves its value $X^{i'}$ in $j$'s memory. Notice that since $i$ is finished computation it does not need to pass its outdated model to $j$, but we assume the worst case.

We proceed by proving the following lemma which upper bounds the expected change in potential:

**Lemma F.1.** *For any time step $t$ we have:*

$$\mathbb{E}[\Gamma_{t+1}] \leq \mathbb{E}[\Gamma_t](1 - \frac{\lambda_2}{2rn}) + (2 + \frac{4r}{\lambda_2})\eta^2 \sum_{i=1}^n \left(4\mathbb{E}\|\widetilde{h}_i(X_t^i)\|^2 + \mathbb{E}\|\widetilde{h}_i\left(X_{p_t^i}^i\right)\|^2\right).$$

*Proof.* First we bound change in potential $\Delta_t = \Gamma_{t+1} - \Gamma_t$ for some fixed time step $t > 0$.

For this, let $\Delta_t^{i,j}$ be the change in potential when we choose agents $(i, j) \in E$ for interaction (While calculating $\Delta_t^{i,j}$ we assume that $X_t$ is fixed). Let $R_t^i = -\eta \widetilde{h}_i(X_t^i) + \dfrac{\eta \widetilde{h}_j\left(X_{p_t^j}^j\right)}{2}$ and $R_t^j =$

$-\eta \widetilde{h}_j(X_t^j) + \frac{\eta \widetilde{h}_i\left(X_{p_t^i}^i\right)}{2}$. We have that:

$$X_{t+1}^i = \frac{X_t^i + X_t^j}{2} + R_t^i.$$

$$X_{t+1}^j = \frac{X_t^i + X_t^j}{2} + R_t^j.$$

$$\mu_{t+1} = \mu_t + \frac{R_t^i + R_t^j}{n}.$$

This gives us that:

$$X_{t+1}^i - \mu_{t+1} = \frac{X_t^i + X_t^j}{2} + \frac{n-1}{n}R_t^i - \frac{1}{n}R_t^j - \mu_t.$$

$$X_{t+1}^i - \mu_{t+1} = \frac{X_t^i + X_t^j}{2} + \frac{n-1}{n}R_t^j - \frac{1}{n}R_t^i - \mu_t.$$

For $k \neq i, j$ we get that

$$X_{t+1}^k - \mu_{t+1} = X_t^k - \frac{1}{n}(R_t^i + R_t^j) - \mu_t.$$

Hence:

$$\begin{aligned}
\Delta_t^{i,j} &= \left\| \frac{X_t^i + X_t^j}{2} + \frac{n-1}{n}R_t^i - \frac{1}{n}R_t^j - \mu_t \right\|^2 - \left\| X_t^i - \mu_t \right\|^2 \\
&\quad + \left\| \frac{X_t^i + X_t^j}{2} + \frac{n-1}{n}R_t^j - \frac{1}{n}R_t^i - \mu_t \right\|^2 - \left\| X_t^j - \mu_t \right\|^2 \\
&\quad + \sum_{k \neq i,j} \left( \left\| X_t^k - \frac{1}{n}(R_t^i + R_t^j) - \mu_t \right\|^2 - \left\| X_t^k - \mu_t \right\|^2 \right) \\
&= 2 \left\| \frac{X_t^i - \mu_t}{2} + \frac{X_t^j - \mu_t}{2} \right\|^2 - \left\| X_t^i - \mu_t \right\|^2 - \left\| X_t^j - \mu_t \right\|^2 \\
&\quad + \left\langle X_t^i - \mu_t + X_t^j - \mu_t, \frac{n-2}{n}R_t^i + \frac{n-2}{n}R_t^j \right\rangle \\
&\quad + \left\| \frac{n-1}{n}R_t^i - \frac{1}{n}R_t^j \right\|^2 + \left\| \frac{n-1}{n}R_t^j - \frac{1}{n}R_t^i \right\|^2 \\
&\quad + \sum_{k \neq i,j} 2\left\langle X_t^k - \mu_t, -\frac{1}{n}(R_t^i + R_t^j) \right\rangle \\
&\quad + \sum_{k \neq i,j} (\frac{1}{n})^2 \| R_t^i + R_t^j \|^2.
\end{aligned}$$

Observe that:

$$\sum_{k=1}^n \left\langle X_t^k - \mu_t, -\frac{1}{n}(R_t^i + R_t^j) \right\rangle = 0.$$

After combining the above two equations, we get that:

$$\begin{aligned}
\Delta_t^{i,j} &= -\frac{\|X_t^i - X_t^j\|^2}{2} + \left\langle X_t^i - \mu_t + X_t^j - \mu_t, R_t^i + R_t^j \right\rangle \\
&\quad + \frac{n-2}{n^2}\left\| R_t^i + R_t^j \right\|^2 + \left\| \frac{n-1}{n}R_t^i - \frac{1}{n}R_t^j \right\|^2 + \left\| \frac{n-1}{n}R_t^j - \frac{1}{n}R_t^i \right\|^2 \\
&\overset{\text{Cauchy-Schwarz}}{\leq} -\frac{\|X_t^i - X_t^j\|^2}{2} + \left\langle X_t^i - \mu_t + X_t^j - \mu_t, R_t^i + R_t^j \right\rangle \\
&\quad + 2\left( \frac{n-2}{n^2} + \frac{1}{n^2} + \frac{(n-1)^2}{n^2} \right)\left( \|R_t^i\|^2 + \|R_t^j\|^2 \right).
\end{aligned} \tag{17}$$

Recall that $R_t^i = -\eta\widetilde{h}_i(X_t^i) + \frac{\eta\widetilde{h}_j\left(X_{p_t^j}^j\right)}{2}$ and $R_t^j = -\eta\widetilde{h}_j(X_t^j) + \frac{\eta\widetilde{h}_i\left(X_{p_t^i}^i\right)}{2}$, Using Cauchy-Schwarz inequality we get that

$$\|R_t^i\|^2 \le 2\eta^2 \|\widetilde{h}_i(X_t^i)\|^2 + \frac{\eta^2}{2}\|\widetilde{h}_j\left(X_{p_t^j}^j\right)\|^2.$$

$$\|R_t^j\|^2 \le 2\eta^2 \|\widetilde{h}_j(X_t^j)\|^2 + \frac{\eta^2}{2}\|\widetilde{h}_i\left(X_{p_t^i}^i\right)\|^2.$$

Denote $2\eta^2\|\widetilde{h}_i(X_t^i)\|^2 + \frac{\eta^2}{2}\|\widetilde{h}_i\left(X_{p_t^i}^i\right)\|^2$ by $S_t^i$ and $2\eta^2\|\widetilde{h}_i(X_t^j)\|^2 + \frac{\eta^2}{2}\|\widetilde{h}_j\left(X_{p_t^j}^j\right)\|^2$ by $S_t^j$. Hence (17) can be rewritten as:

$$
\begin{aligned}
\Delta_t^{i,j} &\le -\frac{\|X_t^i - X_t^j\|^2}{2} + \left\langle X_t^i - \mu_t + X_t^j - \mu_t, R_t^i + R_t^j\right\rangle \\
&\quad + 2\left(\frac{n-2}{n^2} + \frac{1}{n^2} + \frac{(n-1)^2}{n^2}\right)\left(S_t^i + S_t^j\right) \\
&\le -\frac{\|X_t^i - X_t^j\|^2}{2} + \left\langle X_t^i - \mu_t + X_t^j - \mu_t, R_t^i + R_t^j\right\rangle \\
&\quad + 2\left(S_t^i + S_t^j\right).
\end{aligned}
$$

Further:

$$
\begin{aligned}
\left\langle X_t^i - \mu_t + X_t^j - \mu_t, R_t^i + R_t^j\right\rangle &\overset{\text{Young}}{\le} \frac{\lambda_2\left\|X_t^i - \mu_t + X_t^j - \mu_t\right\|^2}{8r} + \frac{2r\left\|R_t^i + R_t^j\right\|^2}{\lambda_2} \\
&\overset{\text{Cauchy-Schwarz}}{\le} \frac{\lambda_2\left\|X_t^i - \mu_t\right\|^2 + \lambda_2\left\|X_t^j - \mu_t\right\|^2}{4r} + \frac{4r\left\|R_t^i\right\|^2 + 4r\left\|R_t^j\right\|^2}{\lambda_2} \\
&\le \frac{\lambda_2\left\|X_t^i - \mu_t\right\|^2 + \lambda_2\left\|X_t^j - \mu_t\right\|^2}{4r} + \frac{4r(S_t^i + S_t^j)}{\lambda_2}.
\end{aligned}
$$

This gives us:

$$
\begin{aligned}
\sum_{(i,j)\in E} \Delta_t^{i,j} &\le \sum_{(i,j)\in E}\left(-\frac{\|X_t^i - X_t^j\|^2}{2} + \frac{\lambda_2\left\|X_t^i - \mu_t\right\|^2 + \lambda_2\left\|X_t^j - \mu_t\right\|^2}{4r} + \frac{4r(S_t^i + S_t^j)}{\lambda_2}\right. \\
&\quad \left. + 2(S_t^i + S_t^j)\right) \\
&\overset{\text{Lemma C.2}}{\le} -\frac{\lambda_2\Gamma_t}{2} + \sum_{i=1}^n (2r + \frac{4r^2}{\lambda_2})S_t^i + \sum_{i=1}^n \frac{\lambda_2\left\|X_t^i - \mu_t\right\|^2}{4} \\
&= -\frac{\lambda_2\Gamma_t}{4} + \sum_{i=1}^n (2r + \frac{4r^2}{\lambda_2})S_t^i.
\end{aligned}
$$

Next, we use the above inequality to upper bound $\Delta_t$ in expectation:

$$
\begin{aligned}
\mathbb{E}[\Delta_t | X_0, X_1, ..., X_t] &= \frac{1}{rn/2}\sum_{(i,j)\in E}\mathbb{E}[\Delta_t^{i,j}|X_0, X_1, ..., X_t] \\
&\le \frac{1}{rn/2}\left(-\frac{\lambda_2\Gamma_t}{4} + \sum_{i=1}^n (2r + \frac{4r^2}{\lambda_2})\mathbb{E}\left[S_t^i|X_0, X_1, ..., X_t\right]\right) \\
&= -\frac{\lambda_2\Gamma_t}{2rn} + \sum_{i=1}^n (4 + \frac{8r}{\lambda_2})\frac{\mathbb{E}\left[S_t^i|X_0, X_1, ..., X_t\right]}{n}.
\end{aligned}
$$

Finally, we remove the conditioning:

$$\mathbb{E}[\Delta_t] = \mathbb{E}[\mathbb{E}[\Delta_t | X_0, X_1, ..., X_t]] \leq -\frac{\lambda_2 \mathbb{E}[\Gamma_t]}{2rn} + (4 + \frac{8r}{\lambda_2}) \sum_{i=1}^{n} \frac{\mathbb{E}[S_t^i]}{n}.$$

By considering the definition of $\Delta_t$ and $S_t^i$, we get the proof of the lemma. $\qquad\square$

Next, we upper bound the second moment of local updates , for any step $t$ and node $i$:

**Lemma F.2.**

$$\sum_{i=1}^{n} \mathbb{E}\|\eta \widetilde{h}_i(X_t^i)\|^2 \leq 2\eta^2 n H^2 M^2.$$

*Proof.*

$$\sum_{i=1}^{n} \mathbb{E}\|\eta \widetilde{h}_i(X_t^i)\|^2 = \eta^2 \sum_{u=1}^{\infty} Pr[H_t^i = u] \sum_{i=1}^{n} \mathbb{E}\|\sum_{q=1}^{u} \widetilde{h}_i^q(X_t^i)\|^2$$

$$\leq \eta^2 \sum_{u=1}^{\infty} Pr[H_t^i = u] \sum_{i=1}^{n} u \sum_{q=1}^{u} \mathbb{E}\|\widetilde{h}_i^q(X_t^i)\|^2$$

$$\overset{(5)}{\leq} \eta^2 \sum_{u=1}^{\infty} Pr[H_t^i = u] u^2 \sum_{i=1}^{n} M^2 \leq 2n\eta^2 H^2 M^2.$$

Where in the last step we used

$$\sum_{u=1}^{\infty} Pr[H_t^i = u] u^2 = \mathbb{E}[(H_t^i)^2] = 2H^2 - H \leq 2H^2.$$

$\qquad\square$

This allows us to upper bound the potential in expectation for any step $t$.

**Lemma F.3.**

$$\mathbb{E}[\Gamma_t] \leq (\frac{40r}{\lambda_2} + \frac{80r^2}{\lambda_2^2}) n\eta^2 H^2 M^2. \tag{18}$$

*Proof.* We prove by using induction. Base case $t = 0$ trivially holds. For an induction step step we assume that $\mathbb{E}[\Gamma_t] \leq (\frac{40r}{\lambda_2} + \frac{80r^2}{\lambda_2^2}) n\eta^2 H^2 M^2 r^2$. We get that :

$$\mathbb{E}[\Gamma_{t+1}] \leq \mathbb{E}[\Gamma_t](1 - \frac{\lambda_2}{2rn}) + (2 + \frac{4r}{\lambda_2})\eta^2 \sum_{i=1}^{n} \left(4\mathbb{E}\|\widetilde{h}_i(X_t^i)\|^2 + \mathbb{E}\|\widetilde{h}_i\left(X_{p_t^i}^i\right)\|^2\right)$$

$$\overset{\text{Lemma (F.2)}}{\leq} (1 - \frac{\lambda_2}{2rn})\mathbb{E}[\Gamma_t] + (20 + \frac{40r}{\lambda_2}) H^2 M^2 \eta^2$$

$$\leq (1 - \frac{\lambda_2}{2rn})(\frac{40r}{\lambda_2} + \frac{80r^2}{\lambda_2^2}) n\eta^2 H^2 M^2 + (20 + \frac{40}{\lambda_2}) H^2 M^2 \eta^2$$

$$= (\frac{40r}{\lambda_2} + \frac{80r^2}{\lambda_2^2}) n\eta^2 H^2 M^2.$$

$\qquad\square$

The next lemma allows us to upper bound $\sum_{i=1}^{n} \mathbb{E}\langle \nabla f(\mu_t), -\widetilde{h}_i(X_t^i)\rangle$ which will be used later once we apply $L$-smoothness to upper bound $f(\mu_{t+1})$. The intuition is as follows: if for each $i$, $\widetilde{h}_i(X_t^i)$ was just a sum of single stochastic gradient($H_i = 1$) by the unbiasedness property we would have to upper bound $\sum_{i=1}^{n} \mathbb{E}\langle \nabla f(\mu_t), -\nabla f(X_t^i)\rangle = \sum_{i=1}^{n} \left(\mathbb{E}\langle \nabla f(\mu_t), \nabla f(\mu_t) - \nabla f(X_t^i)\rangle - \mathbb{E}\|\nabla f(\mu_t)\|^2\right)$, which can be done by using $L$-smoothness and then definition of $\Gamma_t$.

**Lemma F.4.** *For any time step t.*

$$\sum_{i=1}^{n} \mathbb{E}\langle \nabla f(\mu_t), -\widetilde{h}_i(X_t^i)\rangle \leq 2HL^2\mathbb{E}[\Gamma_t] - \frac{3Hn}{4}\mathbb{E}\|\nabla f(\mu_t)\|^2 + 12H^3nL^2M^2\eta^2. \quad (19)$$

*Proof.*

$$\sum_{i=1}^{n}\mathbb{E}\langle \nabla f(\mu_t), -\widetilde{h}_i(X_t^i)\rangle = \sum_{i=1}^{n}\sum_{u=1}^{\infty} Pr[H_t^i = u]\mathbb{E}\langle \nabla f(\mu_t), -\sum_{q=1}^{u}\widetilde{h}_i^q(X_t^i)\rangle$$

$$= \sum_{i=1}^{n}\sum_{u=1}^{\infty} Pr[H_t^i = u]\sum_{q=1}^{u}\left(\mathbb{E}\langle \nabla f(\mu_t), \nabla f(\mu_t) - h_i^q(X_t^i)\rangle - \mathbb{E}\|\nabla f(\mu_t)\|^2\right)$$

$$= \sum_{i=1}^{n}\sum_{u=1}^{\infty} Pr[H_t^i = u]\sum_{q=1}^{u}\left(\mathbb{E}\langle \nabla f(\mu_t), \nabla f(\mu_t) - \nabla f(X_t^i - \sum_{s=0}^{q-1}\eta\widetilde{h}_i^s(X_t^i))\rangle - \mathbb{E}\|\nabla f(\mu_t)\|^2\right)$$

Using Young's inequality we can upper bound $\mathbb{E}\langle \nabla f(\mu_t), \nabla f(\mu_t) - \nabla f(X_t^i - \sum_{s=0}^{q-1}\eta\widetilde{h}_i^s(X_t^i))\rangle$ by $\frac{\mathbb{E}\|\nabla f(\mu_t)\|^2}{4} + \mathbb{E}\left\|\nabla f(\mu_t) - \nabla f(X_t^i - \sum_{s=0}^{q-1}\eta\widetilde{h}_i^s(X_t^i))\right\|^2$. Plugging this in the above inequality we get:

$$\sum_{i=1}^{n}\mathbb{E}\langle \nabla f(\mu_t), -\widetilde{h}_i(X_t^i)\rangle \leq$$

$$\leq \sum_{i=1}^{n}\sum_{u=1}^{\infty} Pr[H_t^i = u]\sum_{q=1}^{u}\left(\mathbb{E}\|\nabla f(\mu_t) - \nabla f(X_t^i - \sum_{s=0}^{q-1}\eta\widetilde{h}_i^s(X_t^i))\|^2 - \frac{3\mathbb{E}\|\nabla f(\mu_t)\|^2}{4}\right)$$

$$\overset{(4)}{\leq} \sum_{i=1}^{n}\sum_{u=1}^{\infty} Pr[H_t^i = u]\sum_{q=1}^{u}\left(L^2\mathbb{E}\|\mu_t - X_t^i + \sum_{s=0}^{q-1}\eta\widetilde{h}_i^s(X_t^i)\|^2 - \frac{3\mathbb{E}\|\nabla f(\mu_t)\|^2}{4}\right).$$

Next we use Cauchy-Schwarz inequality on $\mathbb{E}\|\mu_t - X_t^i + \sum_{s=0}^{q-1}\eta\widetilde{h}_i^s(X_t^i)\|^2$

$$\sum_{i=1}^{n}\mathbb{E}\langle \nabla f(\mu_t), -\widetilde{h}_i(X_t^i)\rangle \leq$$

$$\leq \sum_{i=1}^{n}\sum_{u=1}^{\infty} Pr[H_t^i = u]\sum_{q=1}^{u}\left(2L^2\mathbb{E}\|\mu_t - X_t^i\|^2 + 2L^2\mathbb{E}\|\sum_{s=0}^{q-1}\eta\widetilde{h}_i^s(X_t^i)\|^2 - \frac{3\mathbb{E}\|\nabla f(\mu_t)\|^2}{4}\right)$$

Term $\mathbb{E}\|\sum_{s=0}^{q-1}\eta\widetilde{h}_i^s(X_t^i)\|^2$ can be upper bounded by $q^2M^2$ using Cauchy-Schwarz and assumption (5). Hence:

$$\sum_{i=1}^{n}\mathbb{E}\langle \nabla f(\mu_t), -\widetilde{h}_i(X_t^i)\rangle \leq$$

$$\leq \sum_{i=1}^{n}\sum_{u=1}^{\infty} Pr[H_t^i = u]\sum_{q=1}^{u}\left(2L^2\mathbb{E}\|\mu_t - X_t^i\|^2 + 2L^2\eta^2q^2M^2 - \frac{3\mathbb{E}\|\nabla f(\mu_t)\|^2}{4}\right)$$

$$= \sum_{i=1}^{n}\sum_{u=1}^{\infty} Pr[H_t^i = u]u\left(2L^2\mathbb{E}\|\mu_t - X_t^i\|^2 - \frac{3\mathbb{E}\|\nabla f(\mu_t)\|^2}{4}\right)$$

$$+ \sum_{i=1}^{n}\sum_{u=1}^{\infty} Pr[H_t^i = u]u(u+1)(2u+1)L^2M^2\eta^2/3 \quad (20)$$

Note that:

$$\sum_{i=1}^{n}\sum_{u=1}^{\infty} Pr[H_t^i = u]u\left(2L^2\mathbb{E}\|\mu_t - X_t^i\|^2 - \frac{3\mathbb{E}\|\nabla f(\mu_t)\|^2}{4}\right) = 2HL^2\mathbb{E}[\Gamma_t] - \frac{3Hn}{4}\mathbb{E}\|\nabla f(\mu_t)\|^2.$$

$$(21)$$

Also:

$$\sum_{i=1}^{n}\sum_{u=1}^{\infty} Pr[H_t^i = u]u(u+1)(2u+1)L^2M^2\eta^2/3$$

$$\leq \sum_{i=1}^{n}\sum_{u=1}^{\infty} Pr[H_t^i = u]2u^3L^2M^2\eta^2$$

$$\leq 12H^3nL^2M^2\eta^2. \tag{22}$$

Where in the last step we used (Recall that $H_t^i$ is a geometric random variable with mean $H$):

$$\sum_{u=1}^{\infty} Pr[H_t^i = u]u^3 = \mathbb{E}[(H_t^i)^3] = 6H^3 - 6H^2 + H \leq 6H^3.$$

By plugging inequalities (22) and (21) into inequality (20) we get the proof of the lemma. □

Our next goal is to upper bound $\sum_{i=1}^{n} \mathbb{E}\langle \nabla f(\mu_t), -\widetilde{h}_i(X_{p_t^i}^i)\rangle$.

**Lemma F.5.**

$$\sum_{i=1}^{n}\mathbb{E}\langle \nabla f(\mu_t), \widetilde{h}_i(X_{p_t^i}^i)\rangle \leq 2HL^2\sum_{i=1}^{n}\mathbb{E}\|\mu_t - X_{p_t^i}^i\|^2 + \frac{5Hn}{4}\mathbb{E}\|\nabla f(\mu_t)\|^2 + 12H^3nL^2M^2\eta^2. \tag{23}$$

*Proof.* The proof is very similar to the proof of lemma F.4, except that when we subtract and add term $\mathbb{E}\|\nabla f(\mu_t)\|^2$ in the proof it will eventually end up with a positive sign (After using Young's inequality it will have factor of $\frac{1}{4} + 1$ instead of factor of $\frac{1}{4} - 1$) and we have $\sum_{i=1}^{n}\mathbb{E}\|\mu_t - X_{p_t^i}^i\|^2$ instead of $\Gamma_t = \sum_{i=1}^{n}\mathbb{E}\|\mu_t - X_t^i\|^2$. Thus, we omit the proof in this case. □

Next step is to upper bound $\sum_{i=1}^{n}\mathbb{E}\|\mu_t - X_{p_t^i}^i\|^2$, for this we will need the following lemma:

**Lemma F.6.** *For any node $i$ and time step $t$,*
$$\mathbb{E}\|\mu_t - \mu_{p_t^i}\|^2 \leq 10\eta^2H^2M^2.$$

*Proof.* Notice that

$$\mathbb{E}\|\mu_t - \mu_{p_t^i}\|^2 = \sum_{t'=0}^{t} Pr[p_t^i = t']\mathbb{E}\left\|\sum_{s=t'}^{t-1}\mu_{s+1} - \mu_s\right\|^2 \leq \sum_{t'=0}^{t}\sum_{s=t'}^{t-1} Pr[p_t^i = t'](t-t')\mathbb{E}\left\|\mu_{s+1} - \mu_s\right\|^2. \tag{24}$$

where we used Cauchy-Schwarz inequality in the lastt step. Fix step $s$. Let $u$ and $v$ be nodes which interact at step $s+1$. We have that

$$\mathbb{E}\|\mu_{s+1} - \mu_s\|^2 = \mathbb{E}\left\| -\frac{\eta\widetilde{h}_u(X_s^u)}{n} + \frac{\eta\widetilde{h}_u\left(X_{p_s^u}^u\right)}{2n} - \frac{\eta\widetilde{h}_v(X_s^v)}{n} + \frac{\eta\widetilde{h}_v\left(X_{p_s^v}^v\right)}{2n}\right\|^2$$

$$\leq \frac{4\eta^2}{n^2}\mathbb{E}\|\widetilde{h}_u(X_s^u)\|^2 + \frac{4\eta^2}{n^2}\mathbb{E}\|\widetilde{h}_v(X_s^v)\|^2 +$$

$$+ \frac{\eta^2}{n^2}\mathbb{E}\|\widetilde{h}_u(X_{p_s^u}^u)\|^2 + \frac{\eta^2}{n^2}\mathbb{E}\|\widetilde{h}_v(X_{p_s^v}^v)\|^2.$$

We again used the Cauchy-Schwarz inequality since the expectation is taken only over the randomness of sampling and number of local steps. We can use the approach from lemma $F.2$ to upper bound $E\|\widetilde{h}_u(X_s^u)\|^2$, $\mathbb{E}\|\widetilde{h}_v(X_s^v)\|^2$, $\mathbb{E}\|\widetilde{h}_u(X_{p_s^u}^u)\|^2$ and $\mathbb{E}\|\widetilde{h}_v(X_{p_s^v}^v)\|^2$.

(In the lemma we upper bound the sum of $n$ similar terms, but with $\eta^2$.) Hence:

$$\mathbb{E}\|\mu_t - \mu_{p_t^i}\|^2 \leq \sum_{t'=0}^{t}\sum_{s=t'}^{t-1} Pr[p_t^i = t'](t-t')\mathbb{E}\left\|\mu_{s+1} - \mu_s\right\|^2$$

$$\leq \sum_{t'=0}^{t}\sum_{s=t'}^{t-1} Pr[p_t^i = t'](t-t')^2\frac{20\eta^2H^2M^2}{n^2} = \frac{20\eta^2H^2M^2}{n^2}\mathbb{E}[(p_t^i - t)^2].$$

$t - p_t^i$ is a geometric random variable with mean $n/2$ (because the probability that node $i$ interacts is $2/n$ at every step). Thus, $\mathbb{E}[(p_t^i - t)^2] = 2(\mathbb{E}[t - p_t^i])^2 - \mathbb{E}[t - p_t^i] \leq \frac{n^2}{2}$. Thus, $\mathbb{E}\|\mu_t - \mu_{p_t^i}\|^2 \leq 10\eta^2 H^2 M^2$. $\qquad\square$

Finally we can show that:

**Lemma F.7.** *For any node $i$ and time step $t$,*

$$\sum_{i=1}^{n} \mathbb{E}\|\mu_t - X_{p_t^i}i\|^2 \leq 20nH^2 M^2\eta^2 + (\frac{80r}{\lambda_2} + \frac{160r^2}{\lambda_2^2})n^2\eta^2 H^2 M^2.$$

*Proof.* Using Cauchy-Schwarz inequality we get:

$$\sum_{i=1}^{n}\mathbb{E}\|\mu_t - X_{p_t^i}^i\|^2 \leq \sum_{i=1}^{n}(2\mathbb{E}\|\mu_t - \mu_{p_t^i}\|^2 + 2\mathbb{E}[\mu_{p_t^i} - X_{p_t^i}^i\|^2)$$

$$\leq \sum_{i=1}^{n}(2\mathbb{E}\|\mu_t - \mu_{p_t^i}\|^2 + 2\mathbb{E}[\Gamma_{p_t^i}]) \leq 20nH^2 M^2\eta^2 + (\frac{80r}{\lambda_2} + \frac{160r^2}{\lambda_2^2})n^2\eta^2 H^2 M^2.$$

Where the last inequality comes from Lemmas F.3 and F.6. $\qquad\square$

Now we are ready to prove the main theorem.

**Theorem F.8.** *Let $f$ be an non-convex, $L$-smooth, function satisfying assumption 5, whose minimum $x^\star$ we are trying to find via the non-blocking version of SwarmSGD procedure (See, algorithm 2). Let the number of local stochastic gradient steps performed by each agent upon interaction be a geometric random variable with mean $H$. Let the learning rate we use be $\eta = n/\sqrt{T}$. Define $\mu_t = \sum_{i=1}^{n} X_t^i/n$, where $X_t^i$ is the value of model $i$ after $t$ interactions. Then, for learning rate $\eta = n/\sqrt{T}$ and any $T \geq n^4(n+1)^2$:*

$$\frac{1}{T}\sum_{t=0}^{T-1}\mathbb{E}\|\nabla f(\mu_t)\|^2 \leq \frac{4(f(\mu_0) - f(x^*))}{\sqrt{T}H} + \frac{2304H^2 \max(1, L^2)M^2}{\sqrt{T}}(\frac{r^2}{\lambda_2^2} + 1).$$

*Proof.* Let $\mathbb{E}_t$ denote expectation conditioned on $\{X_1^t, X_2^t, ..., X_n^t\}$. By $L$-smoothness we have that

$$\mathbb{E}_t[f(\mu_{t+1})] \leq f(\mu_t) + \mathbb{E}_t\langle\nabla f(\mu_t), \mu_{t+1} - \mu_t\rangle + \frac{L}{2}\mathbb{E}_t\|\mu_{t+1} - \mu_t\|^2.$$

After removing conditioning:

$$\mathbb{E}[f(\mu_{t+1})] = \mathbb{E}[\mathbb{E}_t[f(\mu_{t+1})]] \leq \mathbb{E}[f(\mu_t)] + \mathbb{E}\langle\nabla f(\mu_t), \mu_{t+1} - \mu_t\rangle + \frac{L}{2}\mathbb{E}\|\mu_{t+1} - \mu_t\|^2. \quad (25)$$

First we look at $\mathbb{E}[\mu_{t+1} - \mu_t]$. If agents $i$ and $j$ interact, (which happens with probability $\frac{1}{rn/2}$). We have that $\mu_{t+1} - \mu_t = -\frac{\eta}{n}\widetilde{h}_i(X_t^i) - \frac{\eta}{n}\widetilde{h}_j(X_t^j) + \frac{\eta}{2n}\widetilde{h}_i(X_{p_t^i}^i) + \frac{\eta}{2n}\widetilde{h}_j(X_{p_t^j}^j)$. Hence we get that

$$\mathbb{E}_t[\mu_{t+1} - \mu_t] = \frac{1}{rn/2}\sum_{(i,j)\in E}\left(\mathbb{E}_t[-\frac{\eta}{n}\widetilde{h}_i(X_t^i) - \frac{\eta}{n}\widetilde{h}_j(X_t^j)] + \frac{\eta}{2n}\widetilde{h}_i(X_{p_t^i}^i) + \frac{\eta}{2n}\widetilde{h}_j(X_{p_t^j}^j).\right)$$

$$= -\frac{2\eta}{n^2}\sum_{i=1}^{n}\mathbb{E}_t[\widetilde{h}_i(X_t^i)] + \frac{\eta}{n^2}\sum_{i=1}^{n}\widetilde{h}_i(X_{p_t^i}^i).$$

and

$$\mathbb{E}_[\mu_{t+1} - \mu_t] = \mathbb{E}[\mathbb{E}_t[\mu_{t+1} - \mu_t]] = -\frac{2\eta}{n^2}\sum_{i=1}^{n}\mathbb{E}[\widetilde{h}_i(X_t^i)] + \frac{\eta}{n^2}\sum_{i=1}^{n}\mathbb{E}[\widetilde{h}_i(X_{p_t^i}^i)].$$

Next we look at $\mathbb{E}\|\mu_{t+1} - \mu_t\|^2$. If agents $i$ and $j$ interact, (which happens with probability $\frac{1}{rn/2}$). We have that $\mu_{t+1} - \mu_t = -\frac{\eta}{n}\widetilde{h}_i(X_t^i) - \frac{\eta}{n}\widetilde{h}_j(X_t^j) + \frac{\eta}{2n}\widetilde{h}_i(X_{p_t^i}^i) + \frac{\eta}{2n}\widetilde{h}_j(X_{p_t^j}^j)$. Hence we get that

$$
\mathbb{E}_t\|\mu_{t+1} - \mu_t\|^2 = \frac{1}{rn/2}\sum_{(i,j)\in E}\mathbb{E}_t\Big\|-\frac{\eta}{n}\widetilde{h}_i(X_t^i) - \frac{\eta}{n}\widetilde{h}_j(X_t^j) + \frac{\eta}{2n}\widetilde{h}_i(X_{p_t^i}^i) + \frac{\eta}{2n}\widetilde{h}_j(X_{p_t^j}^j)\Big\|^2
$$

$$
\overset{Cauchy-Schwarz}{\leq} \frac{1}{rn/2}\sum_{(i,j)\in E}\frac{\eta^2}{n^2}\Big(4\mathbb{E}_t\|\widetilde{h}_i(X_t^i)\|^2 + 4\mathbb{E}_t\|\frac{\eta}{n}\widetilde{h}_j(X_t^j)\|^2
$$

$$
+ \|\widetilde{h}_i(X_{p_t^i}^i)\|^2 + \|\widetilde{h}_j(X_{p_t^j}^j)\|^2\Big)
$$

$$
= \frac{2}{n}\sum_{i=1}^n\frac{4\eta^2}{n^2}\|\widetilde{h}_i(X_t^i)\|^2 + \frac{2}{n}\sum_{i=1}^n\frac{\eta^2}{n^2}\|\widetilde{h}_i(X_{p_t}^i)\|^2
$$

$$
\overset{\text{Lemma F.2}}{\leq} \frac{16\eta^2 H^2 M^2}{n^2} + \frac{2}{n}\sum_{i=1}^n\frac{\eta^2}{n^2}\|\widetilde{h}_i(X_{p_t}^i)\|^2.
$$

and

$$
\mathbb{E}\|\mu_{t+1} - \mu_t\|^2 = \mathbb{E}[[\mathbb{E}_t\|\mu_{t+1} - \mu_t\|^2]]
$$

$$
\leq \frac{16\eta^2 H^2 M^2}{n^2} + \frac{2}{n}\sum_{i=1}^n\frac{\eta^2}{n^2}\mathbb{E}\|\widetilde{h}_i(X_{p_t}^i)\|^2 \overset{\text{Lemma F.2}}{\leq} \frac{20\eta^2 H^2 M^2}{n^2}.
$$

Hence, we can rewrite (25) as:

$$
\mathbb{E}[f(\mu_{t+1})] \leq \mathbb{E}[f(\mu_t)] + \frac{2\eta}{n^2}\sum_{i=1}^n\mathbb{E}\Big\langle\nabla f(\mu_t), -\widetilde{h}_i(X_t^i)\Big\rangle + \frac{\eta}{n^2}\sum_{i=1}^n\mathbb{E}\Big\langle\nabla f(\mu_t), \widetilde{h}_i(X_{p_t^i}^i)\Big\rangle
$$

$$
+ \frac{20L\eta^2 H^2 M^2}{n^2}.
$$

Next, we use Lemmas F.4 and F.5:

$$
\mathbb{E}[f(\mu_{t+1})] \leq \mathbb{E}[f(\mu_t)] + \frac{2\eta}{n^2}\Big(2HL^2\mathbb{E}[\Gamma_t] - \frac{3Hn}{4}\mathbb{E}\|\nabla f(\mu_t)\|^2 + 12H^3nL^2M^2\eta^2\Big)
$$

$$
+ \frac{\eta}{n^2}\Big(2HL^2\sum_{i=1}^n\mathbb{E}\|\mu_t - X_{p_t^i}^i\|^2 + \frac{5Hn}{4}\mathbb{E}\|\nabla f(\mu_t)\|^2 + 12H^3nL^2M^2\eta^2\Big)
$$

$$
+ \frac{4L\eta^2 H^2 M^2}{n^2}
$$

$$
= \mathbb{E}[f(\mu_t)] + \frac{4HL^2\eta\mathbb{E}[\Gamma_t]}{n^2} + \frac{2HL^2\eta\Big(\sum_{i=1}^n\mathbb{E}\|\mu_t - X_{p_t^i}^i\|^2\Big)}{n^2} - \frac{H\eta}{4n}\mathbb{E}\|\nabla f(\mu_t)\|^2
$$

$$
+ \frac{36H^3L^2M^2\eta^3}{n} + \frac{20L\eta^2 H^2 M^2}{n^2}.
$$

We use Lemmas F.3 and F.7 to upper bound $\mathbb{E}[\Gamma_t]$ and $\sum_{i=1}^n\mathbb{E}\|\mu_t - X_{p_t^i}^i\|^2$ respectively :

$$
\mathbb{E}[f(\mu_{t+1})] - \mathbb{E}[f(\mu_t)] \leq \Big(\frac{160r}{\lambda_2} + \frac{320r^2}{\lambda_2^2}\Big)\frac{\eta^3 H^3 M^2 L^2(n^2 + n)}{n^2} - \frac{Hn}{4}\mathbb{E}\|\nabla f(\mu_t)\|^2
$$

$$
+ \frac{76H^3L^2M^2\eta^3}{n} + \frac{20L\eta^2 H^2 M^2}{n^2}.
$$

by summing the above inequality for $t = 0$ to $t = T - 1$, we get that

$$
\mathbb{E}[f(\mu_T)] - f(\mu_0) \leq \sum_{t=0}^{T-1}\Big(\Big(\frac{160r}{\lambda_2} + \frac{320r^2}{\lambda_2^2}\Big)\frac{\eta^3 H^3 M^2 L^2(n+1)}{n} - \frac{\eta H}{4n}\mathbb{E}\|\nabla f(\mu_t)\|^2
$$

$$
+ \frac{20L\eta^2 H^2 M^2}{n^2} + \frac{76H^3L^2M^2\eta^3}{n}\Big).
$$

From this we get that :

$$\sum_{t=0}^{T-1}\frac{\eta H}{4n}\mathbb{E}\|\nabla f(\mu_t)\|^2 \leq f(\mu_0) - \mathbb{E}[f(\mu_T)] + (\frac{160r}{\lambda_2} + \frac{320r^2}{\lambda_2^2})\frac{\eta^3 H^3 M^2 L^2 T(n+1)}{n}$$
$$+ \frac{20TL\eta^2 H^2 M^2}{n^2} + \frac{76TH^3 L^2 M^2 \eta^3}{n}.$$

Note that $\mathbb{E}[f(\mu_T)] \geq f(x^*)$, hence after multiplying the above inequality by $\frac{4n}{\eta HT}$ we get that

$$\frac{1}{T}\sum_{t=0}^{T-1}\mathbb{E}\|\nabla f(\mu_t)\|^2 \leq \frac{4n(f(\mu_0) - f(x^*))}{TH\eta} + (\frac{640r}{\lambda_2} + \frac{1280r^2}{\lambda_2^2})\eta^2 H^2 M^2 L^2 (n+1)$$
$$+ \frac{80L\eta HM^2}{n} + 304H^2 L^2 M^2 \eta^2.$$

Observe that $\eta = n/\sqrt{T} \leq \frac{1}{n(n+1)}$, since $T \geq n^4(n+1)^2$. This allows us to finish the proof:

$$\frac{1}{T}\sum_{t=0}^{T-1}\mathbb{E}\|\nabla f(\mu_t)\|^2 \leq \frac{4n(f(\mu_0) - f(x^*))}{TH\eta} + (\frac{640r}{\lambda_2} + \frac{1280r^2}{\lambda_2^2})\frac{L^2 \eta M^2 H^2}{n}$$
$$+ \frac{80L\eta HM^2}{n} + \frac{304H^2 L^2 M^2 \eta}{n}$$
$$= \frac{(4f(\mu_0) - f(x^*))}{\sqrt{T}H} + (\frac{640r}{\lambda_2} + \frac{1280r^2}{\lambda_2^2})\frac{L^2 M^2 H^2}{\sqrt{T}}$$
$$+ \frac{80LHM^2}{\sqrt{T}} + \frac{304H^2 L^2 M^2}{\sqrt{T}}$$
$$\leq \frac{4(f(\mu_0) - f(x^*))}{\sqrt{T}H} + \frac{2304H^2 \max(1, L^2)M^2}{\sqrt{T}}(\frac{r^2}{\lambda_2^2} + 1).$$

$\square$

## G  ANALYSIS OF QUANTIZED AVERAGING, ASSUMING SECOND MOMENT BOUND AND RANDOM NUMBER OF LOCAL STEPS

First we define how quantization of models changes our interactions. Both the algorithm and the analysis in this case are similar to those of Section F. Let $i$ and $j$ be nodes which interact at step $t + 1$, we set

$$X_{t+1/2}^i = \frac{X_t^i}{2} + \frac{X_t^{j'}}{2},$$
$$X_{t+1/2}^j = \frac{X_t^j}{2} + \frac{X_t^{i'}}{2},$$

and

$$X_{t+1}^i = X_{t+1/2}^i - \eta\widetilde{h}_i(X_t^i),$$
$$X_{t+1}^j = X_{t+1/2}^j - \eta\widetilde{h}_j(X_t^j),$$

where for each node $k$, $X_t^{k'}$ is a quantized version of the model $X_t^k$. We use the quantization provided in Davies et al. (2020). The key property of this quantization scheme is summarized below:

**Lemma G.1.** *Let $q$ be a parameter we will fix later. If the inputs $x^u$, $x^v$ at nodes $u$ and $v$, respectively satisfy $\|x^u - x^v\| \leq q^{q^d}\epsilon$, then with probability at least $1 - \log\log(\frac{1}{\epsilon}\|x^u - x^v\|) \cdot O(q^{-d})$, the quantization algorithm of Davies et al. (2020) provides node $v$ with an unbiased estimate $x^{u'}$ of $x^u$, with $\|x^{u'} - x^u\| \leq (q^2 + 7)\epsilon$, and uses $O\left(d\log(\frac{q}{\epsilon}\|x^u - x^v\|)\right)$ bits to do so.*

For our purposes, $x^u$ and $x^v$ are the local models of the nodes $u$ and $v$ and $d$ is their dimension (we omit the time step here). In the following , we refer to the above lemma as the quantization lemma. Recall that in section E, for each node $k$, if $p_t^k + 1$ is the last time interacting before and including step $t$:

$$X_t^{k'} = X_{p_t^k+1/2}^i = X_t^k + \eta \widetilde{h}_k(X_{p_t^k}^k). \tag{26}$$

**Analysis Outline.** The crucial property we used in the analysis is that $\mathbb{E}[\widetilde{h}_k(X_{p_t^k}^k)] \leq 2H^2 M^2 \eta^2$ (see Lemma F.2). We also used that for $\widetilde{h}_k(X_{p_t^k}^k)$ , we can use the smoothness property (4) in Lemmas F.5 and F.7. In our case we plan to use the quantization lemma above. For this, first notice that the estimate is unbiased: this means that $\mathbb{E}[X_t^{k'}] = X_t^k$, eliminating the need to use Lemma F.5 and subsequently F.7, since $\sum_{i=1}^n \mathbb{E}\langle \nabla f(\mu_t), X_t^i - X_t^{i'} \rangle = 0$ in our case. Secondly if we set $(q^2 + 7)\epsilon = H\eta M$ we also satisfy Lemma F.2. This means that entire analysis can be replicated, and even further as in the case of section E we will only need $T \geq n^4$ (In one case we do not use Lemma F.5 at all, and in the second case upper bound can be replaced by 0, which is the same as not using it). Now we concentrate on calculating the probability that $\|x^u - x^v\| \leq q^{q^d}\epsilon$ (which we call the distance criterion) required by the quantization lemma to hold over $T$ steps and each pair of nodes. We also need to calculate the probability with which we fail to decode.

Assume that Lemma F.3 holds for step $t$, as in the proof of this lemma we will use induction and Lemma F.1 (we omit the proof since it will be exactly the same given that the conditions we discuss above hold). That is: $\mathbb{E}[\Gamma_t] \leq (\frac{40r}{\lambda_2} + \frac{80r^2}{\lambda_2^2})n\eta^2 H^2 M^2$.

Notice that for a pair of nodes $u$ and $v$, $\|X_t^u - X_t^v\|^2 \leq 2\Gamma_t$ (Using Cauchy-Schwarz). Hence we need to calculate the probability that $\Gamma_t \geq (q^{q^d}\epsilon)^2/2$. Using Markov's inequality, the probability of this happening is at most:

$$\frac{2\mathbb{E}[\Gamma_t]}{(q^{q^d}\epsilon)^2} \leq \left(\frac{80r}{\lambda_2} + \frac{160r^2}{\lambda_2^2}\right)\frac{n(q^2+7)^2\epsilon^2}{(q^{q^d}\epsilon)^2} = \left(\frac{80r}{\lambda_2} + \frac{160r^2}{\lambda_2^2}\right)\frac{n(q^2+7)^2}{(q^{q^d})^2}.$$

We set $q = 2 + T^{3/d}$, this means that $(q^{q^d})^2 \geq 2^{T^3}$. So given that $T \geq n^4$, we have that $Pr[\Gamma_t \geq (q^{q^d}\epsilon)^2/2] \leq O(1/T^4)$ (note that $r \leq n - 1$ and $\lambda_2 = \Omega(1/n^2)$ since our graph is connected). Hence the distance criterion is satisfied with probability $1 - O(1/T^2)$. Given that it is satisfied, we also have failure probability $\log\log(\frac{1}{\epsilon}\|X_t^u - X_t^v\|) \cdot O(q^{-d}) = O(\frac{\log\log T}{T^3})$.

So, the total probability of failure, either due to contravening the distance criterion or by probabilistic failure, is at most $O(T^{-2})$. Hence with probability $1 - O(1/T^2)$ we can use Lemma $F$.1 and prove that

$$\mathbb{E}[\Gamma_{t+1}] \leq \left(\frac{40r}{\lambda_2} + \frac{80r^2}{\lambda_2^2}\right)n\eta^2 H^2 M^2.$$

What is left is to union bound over $T$ steps[2], and we get that with probability $1 - O(1/T) = 1 - O(1/n^4)$ the quantization algorithm never fails and the distance criterion is always satisfied. The total number of bits used per step is $O(d\log q) = O(d + \log T)$.

With this we can state the main theorem:

**Theorem G.2.** *Let $f$ be an non-convex, $L$-smooth function, whose stochastic gradients satisfy the bounded second moment assumption above. Consider the quantized version of the algorithm 1. Let the number of local stochastic gradient steps performed by each agent upon interaction be a geometric random variable with mean $H$. Let the learning rate we use be $\eta = n/\sqrt{T}$. Define $\mu_t = \sum_{i=1}^n X_t^i/n$, where $X_t^i$ is a value of model $i$ after $t$ interactions, be the average of the local parameters. Then, for learning rate $\eta = n/\sqrt{T}$ and any number of interactions $T \geq n^4$, with probability at least $1 - O(1/n^4)$ we have that:*

$$\frac{1}{T}\sum_{t=0}^{T-1}\mathbb{E}\|\nabla f(\mu_t)\|^2 \leq \frac{4(f(\mu_0) - f(x^*))}{\sqrt{T}H} + \frac{2304H^2\max(1,L^2)M^2}{\sqrt{T}}\left(\frac{r^2}{\lambda_2^2} + 1\right).$$

*and additionally we use $O(d + \log T)$ communication bits per step.*

---

[2]Note that we do not need to union bound over all pairs of $u$ and $v$, since can assume that $u$ and $v$ are the ones which interact at step $t + 1$

# H FIXED NUMBER OF LOCAL STEPS WITH VARIANCE BOUND AND NON-IDENTICALLY DISTRIBUTED DATA

We again deal with a non-convex L-smooth function, but we no longer require a second moment bound, and no longer assume that data is distributed identically.

We use a constant learning rate $\eta$ and fixed local steps sizes $H_t^i = H$, for each node $i$ and step $t$.

Each agent $i$ has access to local function $f_i$ such that:

1. For each agent $i$, the gradient $\nabla f_i(x)$ is $L$-Lipschitz continuous for some $L > 0$, i.e. for all $x, y \in \mathbb{R}^d$:
$$\|\nabla f_i(x) - \nabla f_i(y)\| \leq L\|x - y\|. \tag{27}$$

2. for every $x \in \mathbb{R}^d$:
$$\sum_{i=1}^{n} f_i(x)/n = f(x). \tag{28}$$

3. For each agent $i$ and $x \in \mathbb{R}^d$:
$$\mathbb{E}[\widetilde{g}_i(x)] = \nabla f_i(x). \tag{29}$$

4. For each agent $i$ and $x \in \mathbb{R}^d$ there exist $\sigma^2$ such that:
$$\mathbb{E}\|\widetilde{g}_i(x) - \nabla f_i(x)\|^2 \leq \sigma^2. \tag{30}$$

5. For each $x \in \mathbb{R}^d$ there exist $\sigma^2$ such that:
$$\sum_{i=1}^{n} \|\nabla f_i(x) - \nabla f(x)\|^2/n \leq \rho^2. \tag{31}$$

Notice that since data is not distributed identically, for $1 \leq q \leq H$, we no longer have that

$$h_i^q(X_t^i) = \mathbb{E}[\tilde{h}_i^q(X_t^i)] = \mathbb{E}[\widetilde{g}_i(X_t^i - \sum_{s=0}^{q-1} \eta \widetilde{h}_i^s(X_t^i))] = \nabla f(X_t^i - \sum_{s=0}^{q-1} \eta \widetilde{h}_i^s(X_t^i)).$$

Instead,

$$h_i^q(X_t^i) = \mathbb{E}[\tilde{h}_i^q(X_t^i)] = \mathbb{E}[\widetilde{g}_i(X_t^i - \sum_{s=0}^{q-1} \eta \widetilde{h}_i^s(X_t^i))] = \nabla f_i(X_t^i - \sum_{s=0}^{q-1} \eta \widetilde{h}_i^s(X_t^i)).$$

We proceed by proving the following lemma which upper bounds the expected change in potential:

**Lemma H.1.** *For any time step $t$, we have:*

$$\mathbb{E}[\Gamma_{t+1}] \leq (1 - \frac{\lambda_2}{2rn})\mathbb{E}[\Gamma_t] + (2 + \frac{8r}{\lambda_2})\eta^2 \sum_{i=1}^{n} \frac{\mathbb{E}\|\widetilde{h}_i(X_t^i)\|^2}{n}$$

*Proof.* First we bound change in potential $\Delta_t = \Gamma_{t+1} - \Gamma_t$ for some fixed time step $t > 0$.

For this, let $\Delta_t^{i,j}$ be the change in potential when we choose agents $(i, j) \in E$ for interaction.

We have that:

$$X_{t+1}^i = (X_t^i + X_t^j + \eta \widetilde{h}_i(X_t^i) + \eta \widetilde{h}_j(X_t^j))/2.$$
$$X_{t+1}^j = (X_t^i + X_t^j + \eta \widetilde{h}_i(X_t^i) + \eta \widetilde{h}_j(X_t^j))/2.$$
$$\mu_{t+1} = \mu_t + \eta \widetilde{h}_i(X_t^i)/n + \eta \widetilde{h}_j(X_t^j)/n.$$

This gives us that:

$$X_{t+1}^i - \mu_{t+1} = \frac{X_t^i + X_t^j}{2} + \frac{n-2}{2n} \eta \widetilde{h}_i(X_t^i) + \frac{n-2}{2n} \eta \widetilde{h}_j(X_t^j) - \mu_t.$$
$$X_{t+1}^j - \mu_{t+1} = \frac{X_t^i + X_t^j}{2} + \frac{n-2}{2n} \eta \widetilde{h}_i(X_t^i) + \frac{n-2}{2n} \eta \widetilde{h}_j(X_t^j) - \mu_t.$$

For $k \neq i, j$ we get that

$$X_{t+1}^k - \mu_{t+1} = X_t^k - \frac{1}{n}(\eta \widetilde{h}_i(X_t^i) + \eta \widetilde{h}_j(X_t^j)) - \mu_t.$$

Hence:

$$
\begin{aligned}
\Delta_t^{i,j} &= \left\| (X_t^i + X_t^j)/2 + \frac{n-2}{2n}(\eta \widetilde{h}_i(X_t^i) + \eta \widetilde{h}_j(X_t^j)) - \mu_t \right\|^2 - \left\| X_t^i - \mu_t \right\|^2 \\
&\quad + \left\| (X_t^i + X_t^j)/2 + \frac{n-2}{2n}(\eta \widetilde{h}_i(X_t^i) + \eta \widetilde{h}_j(X_t^j)) - \mu_t \right\|^2 - \left\| X_t^j - \mu_t \right\|^2 \\
&\quad + \sum_{k \neq i,j} \left( \left\| X_t^k - \frac{1}{n}(\eta \widetilde{h}_i(X_t^i) + \eta \widetilde{h}_j(X_t^j)) - \mu_t \right\|^2 - \left\| X_t^k - \mu_t \right\|^2 \right) \\
&= 2 \left\| \frac{X_t^i - \mu_t}{2} + \frac{X_t^j - \mu_t}{2} \right\|^2 - \left\| X_t^i - \mu_t \right\|^2 - \left\| X_t^j - \mu_t \right\|^2 \\
&\quad + 2 \left\langle X_t^i - \mu_t + X_t^j - \mu_t, \frac{n-2}{2n}\eta \widetilde{h}_i(X_t^i) + \frac{n-2}{2n}\eta \widetilde{h}_j(X_t^j) \right\rangle \\
&\quad + 2(\frac{n-2}{2n})^2 \| \eta \widetilde{h}_i(X_t^i) + \eta \widetilde{h}_j(X_t^j) \|^2 \\
&\quad + \sum_{k \neq i,j} 2 \left\langle X_t^k - \mu_t, -\frac{1}{n}(\eta \widetilde{h}_i(X_t^i) + \eta \widetilde{h}_j(X_t^j)) \right\rangle \\
&\quad + \sum_{k \neq i,j} (\frac{1}{n})^2 \| \eta \widetilde{h}_i(X_t^i) + \eta \widetilde{h}_j(X_t^j) \|^2.
\end{aligned}
$$

Observe that:

$$\sum_{k=1}^n \left\langle X_t^k - \mu_t, -\frac{1}{n}(\eta \widetilde{h}_i(X_t^i) + \eta \widetilde{h}_j(X_t^j)) \right\rangle = 0.$$

After combining the above two equations, we get that:

$$
\begin{aligned}
\Delta_t^{i,j} &= -\frac{\|X_t^i - X_t^j\|^2}{2} + \left\langle X_t^i - \mu_t + X_t^j - \mu_t, \eta \widetilde{h}_i(X_t^i) + \eta \widetilde{h}_j(X_t^j) \right\rangle \\
&\quad + \left( 2(\frac{n-2}{2n})^2 + (n-2)(\frac{1}{n})^2 \right) \| \eta \widetilde{h}_i(X_t^i) + \eta \widetilde{h}_j(X_t^j) \|^2 \\
&\overset{\text{Cauchy-Schwarz}}{\leq} -\frac{\|X_t^i - X_t^j\|^2}{2} + \left\langle X_t^i - \mu_t + X_t^j - \mu_t, \eta \widetilde{h}_i(X_t^i) + \eta \widetilde{h}_j(X_t^j) \right\rangle \\
&\quad + 2 \left( 2(\frac{n-2}{2n})^2 + (n-2)(\frac{1}{n})^2 \right) \left( \| \eta \widetilde{h}_i(X_t^i) \|^2 + \| \eta \widetilde{h}_j(X_t^j) \|^2 \right) \\
&\leq -\frac{\|X_t^i - X_t^j\|^2}{2} + \left\langle X_t^i - \mu_t + X_t^j - \mu_t, \eta \widetilde{h}_i(X_t^i) + \eta \widetilde{h}_j(X_t^j) \right\rangle \\
&\quad + \left( \| \eta \widetilde{h}_i(X_t^i) \|^2 + \| \eta \widetilde{h}_j(X_t^j) \|^2 \right).
\end{aligned}
$$

This gives us:

$$\sum_{(i,j)\in E} \Delta_t^{i,j} \leq \sum_{(i,j)\in E} \left( -\frac{\|X_t^i - X_t^j\|^2}{2} + \left\langle X_t^i - \mu_t + X_t^j - \mu_t, \eta \widetilde{h}_i(X_t^i) + \eta \widetilde{h}_j(X_t^j) \right\rangle \right.$$
$$\left. + \|\eta \widetilde{h}_i(X_t^i)\|^2 + \|\eta \widetilde{h}_j(X_t^j)\|^2 \right)$$

$$\overset{\text{Lemma C.2}}{\leq} -\frac{\lambda_2 \Gamma_t}{2} + \sum_{i=1}^n r\|\eta \widetilde{h}_i(X_t^i)\|^2$$
$$+ \sum_{(i,j)\in E} \left\langle X_t^i - \mu_t + X_t^j - \mu_t, \eta \widetilde{h}_i(X_t^i) + \eta \widetilde{h}_j(X_t^j) \right\rangle$$

$$\overset{\text{Young}}{\leq} -\frac{\lambda_2 \Gamma_t}{2} + \sum_{i=1}^n r\|\eta \widetilde{h}_i(X_t^i)\|^2$$
$$+ \sum_{(i,j)\in E} \left( \frac{\lambda_2 \left\|X_t^i - \mu_t + X_t^j - \mu_t\right\|^2}{8r} + \frac{2r\left\|\eta \widetilde{h}_i(X_t^i) + \eta \widetilde{h}_j(X_t^j)\right\|^2}{\lambda_2} \right)$$

$$\overset{\text{Cauchy-Schwarz}}{\leq} -\frac{\lambda_2 \Gamma_t}{2} + \sum_{i=1}^n r\|\eta \widetilde{h}_i(X_t^i)\|^2$$
$$+ \sum_{(i,j)\in E} \left( \frac{\lambda_2 \left\|X_t^i - \mu_t\right\|^2 + \lambda_2 \left\|X_t^j - \mu_t\right\|^2}{4r} \right.$$
$$\left. + \frac{4r\left\|\eta \widetilde{h}_i(X_t^i)\right\|^2 + 4r\left\|\eta \widetilde{h}_j(X_t^j)\right\|^2}{\lambda_2} \right)$$

$$= -\frac{\lambda_2 \Gamma_t}{2} + \sum_{i=1}^n r\|\eta \widetilde{h}_i(X_t^i)\|^2 + \sum_{i=1}^n \frac{\lambda_2 \left\|X_t^i - \mu_t\right\|^2}{4}$$
$$+ \sum_{i=1}^n \frac{4r^2 \|\eta \widetilde{h}_i(X_t^i)\|^2}{\lambda_2}$$

Next, we use definition of $\Gamma_t$:

$$\sum_{(i,j)\in E} \Delta_t^{i,j} \leq -\frac{\lambda_2 \Gamma_t}{4} + \sum_{i=1}^n (r + \frac{4r^2}{\lambda_2})\|\eta \widetilde{h}_i(X_t^i)\|^2. \tag{32}$$

Next, we use the above inequality to upper bound $\Delta_t$ in expectation:

$$\mathbb{E}[\Delta_t | X_0, X_1, ..., X_t] = \frac{1}{rn/2} \sum_{(i,j)\in E} \mathbb{E}[\Delta_t^{i,j} | X_0, X_1, ..., X_t]$$

$$\leq \frac{1}{rn/2} \left( -\frac{\lambda_2 \Gamma_t}{4} + \sum_{i=1}^n (r + \frac{4r^2}{\lambda_2})\mathbb{E}\left[ \|\eta \widetilde{h}_i(X_t^i)\|^2 | X_0, X_1, ..., X_t \right] \right)$$

$$= -\frac{\lambda_2 \Gamma_t}{2rn} + \sum_{i=1}^n (2 + \frac{8r}{\lambda_2})\eta^2 \frac{\mathbb{E}\left[ \|\widetilde{h}_i(X_t^i)\|^2 | X_0, X_1, ..., X_t \right]}{n}.$$

Finally, we remove the conditioning:

$$\mathbb{E}[\Delta_t] = \mathbb{E}[\mathbb{E}[\Delta_t | X_0, X_1, ..., X_t]] \leq -\frac{\lambda_2 \mathbb{E}[\Gamma_t]}{2rn} + (2 + \frac{8r}{\lambda_2})\eta^2 \sum_{i=1}^n \frac{\mathbb{E}\|\widetilde{h}_i(X_t^i)\|^2}{n}.$$

By considering the definition of $\Delta_t$, we get the proof of the lemma. $\qquad \square$

**Lemma H.2.** *For any $1 \leq q \leq H$ and step $t$, we have that*

$$\sum_{i=1}^{n} \mathbb{E}\|\nabla f_i(\mu_t) - h_i^q(X_t^i)\|^2 \leq 2L^2 \mathbb{E}[\Gamma_t] + \sum_{i=1}^{n} 2L^2 \eta^2 \mathbb{E}\|\sum_{s=0}^{q-1} \widetilde{h}_i^s(X_t^i)\|^2.$$

*Proof.*

$$\begin{aligned}
\sum_{i=1}^{n} \mathbb{E}\|\nabla f_i(\mu_t) - h_i^q(X_t^i)\|^2 &= \sum_{i=1}^{n} \mathbb{E}\|\nabla f_i(\mu_t) - \nabla f_i(X_t^i - \sum_{s=0}^{q-1} \eta \widetilde{h}_i^s(X_t^i))\|^2 \\
&\overset{(27)}{\leq} \sum_{i=1}^{n} L^2 \mathbb{E}\|\mu_t - X_t^i + \eta \widetilde{h}_i^{q-1}(X_t^i))\|^2 \\
&\overset{Cauchy-Schwarz}{\leq} \sum_{i=1}^{n} 2L^2 \mathbb{E}\|X_t^i - \mu_t\|^2 + \sum_{i=1}^{n} 2L^2 \eta^2 \mathbb{E}\|\sum_{s=0}^{q-1} \widetilde{h}_i^s(X_t^i)\|^2 \\
&= 2L^2 \mathbb{E}[\Gamma_t] + \sum_{i=1}^{n} 2L^2 \eta^2 \mathbb{E}\|\sum_{s=0}^{q-1} \widetilde{h}_i^s(X_t^i)\|^2.
\end{aligned}$$

$\square$

**Lemma H.3.** *For any $1 \leq q \leq H$ and step $T$, we have that*

$$\sum_{i=1}^{n} \mathbb{E}\|\widetilde{h}_i^q(X_t^i)\|^2 \leq n\sigma^2 + 4n\rho^2$$

$$+ 16L^2 \mathbb{E}[\Gamma_t] + \sum_{i=1}^{n} 16L^2 \eta^2 \mathbb{E}\|\sum_{s=0}^{q-1} \widetilde{h}_i^s(X_t^i)\|^2 + 4n\mathbb{E}\|\sum_{i=1}^{n} h_i^q(X_t^i)/n\|^2.$$

*Proof.*

$$\begin{aligned}
\sum_{i=1}^{n} \mathbb{E}\|\widetilde{h}_i^q(X_t^i)\|^2 &\leq \sum_{i=1}^{n} (\sigma^2 + \mathbb{E}\|h_i^q(X_t^i)\|^2) = n\sigma^2 + \sum_{i=1}^{n} \mathbb{E}\|\nabla f_i(X_t^i - \sum_{s=0}^{q-1} \eta \widetilde{h}_i^s(X_t^i))\|^2 \\
&\overset{(28)}{\leq} n\sigma^2 + \sum_{i=1}^{n} \mathbb{E}\Big\| \nabla f_i(X_t^i - \eta \widetilde{h}_i^{q-1}(X_t^i)) - \nabla f_i(\mu_t) + \nabla f_i(\mu_t) - \nabla f(\mu_t) + \sum_{j=1}^{n} \nabla f_j(\mu_t)/n \\
&\qquad\qquad - \sum_{j=1}^{n} \nabla f_j(X_t^j - \sum_{s=0}^{q-1} \eta \widetilde{h}_j^s(X_t^j))/n + \sum_{j=1}^{n} \nabla f_j(X_t^j - \sum_{s=0}^{q-1} \eta \widetilde{h}_j^s(X_t^j))/n \Big\|^2 \\
&\overset{Cauchy-Schwarz}{\leq} n\sigma^2 + \sum_{i=1}^{n} 4\mathbb{E}\|\nabla f_i(\mu_t) - h_i^q(X_t^i)\|^2 + 4n\mathbb{E}\|\sum_{i=1}^{n} (\nabla f_i(\mu_t) - h_i^q(X_t^i))/n\|^2 \\
&\qquad\qquad + 4n\mathbb{E}\|\sum_{i=1}^{n} h_i^q(X_t^i)/n\|^2 + 4\sum_{i=1}^{n} \|\nabla f_i(\mu_t) - \nabla f(\mu_t)\|^2 \\
&\overset{Cauchy-Schwarz,(31)}{\leq} n\sigma^2 + 4n\rho^2 + \sum_{i=1}^{n} 8\mathbb{E}\|\nabla f_i(\mu_t) - h_i^q(X_t^i)\|^2 + 4n\mathbb{E}\|\sum_{i=1}^{n} h_i^q(X_t^i)/n\|^2 \\
&\overset{Lemma\ H.2}{\leq} n\sigma^2 + 4n\rho^2 + 16L^2 \mathbb{E}[\Gamma_t] + \sum_{i=1}^{n} 16L^2 \eta^2 \mathbb{E}\|\sum_{s=0}^{q-1} \widetilde{h}_i^s(X_t^i)\|^2 + 4n\mathbb{E}\|\sum_{i=1}^{n} h_i^q(X_t^i)/n\|^2.
\end{aligned}$$

$\square$

Next we use the above lemma to show the upper bound for $\sum_{q=1}^{H} \sum_{i=1}^{n} \mathbb{E}\|\widetilde{h}_i^q(X_t^i)\|^2$:

**Lemma H.4.** *For $\eta \leq \frac{1}{6LH}$, we have that :*

$$\sum_{q=1}^{H}\sum_{i=1}^{n}\mathbb{E}\|\widetilde{h}_i^q(X_t^i)\|^2 \leq 2Hn(\sigma^2 + 4\rho^2) + 32HL^2\mathbb{E}[\Gamma_t] + 8n\sum_{q=1}^{H}\mathbb{E}\|\sum_{i=1}^{n}h_i^q(X_t^i)/n\|^2$$

*Proof.* Notice that if $\eta \leq \frac{1}{6LH}$ the Lemma H.3 gives us that :

$$\sum_{i=1}^{n}\mathbb{E}\|\widetilde{h}_i^q(X_t^i)\|^2 \leq n(\sigma^2 + 4\rho^2) + 16L^2\mathbb{E}[\Gamma_t] \tag{33}$$

$$+ \sum_{i=1}^{n}\frac{1}{2H^2}\mathbb{E}\|\sum_{s=0}^{q-1}\widetilde{h}_i^s(X_t^i)\|^2 + 4n\mathbb{E}\|\sum_{i=1}^{n}h_i^q(X_t^i)/n\|^2$$

$$\leq n(\sigma^2 + 4\rho^2) + 16L^2\mathbb{E}[\Gamma_t] + \sum_{i=1}^{n}\frac{q}{2H^2}\sum_{s=0}^{q-1}\mathbb{E}\|\widetilde{h}_i^s(X_t^i)\|^2 + 4n\mathbb{E}\|\sum_{i=1}^{n}h_i^q(X_t^i)/n\|^2$$

$$\leq n(\sigma^2 + 4\rho^2) + 16L^2\mathbb{E}[\Gamma_t] + \sum_{i=1}^{n}\frac{1}{2H}\sum_{s=0}^{q-1}\mathbb{E}\|\widetilde{h}_i^s(X_t^i)\|^2 + 4n\mathbb{E}\|\sum_{i=1}^{n}h_i^q(X_t^i)/n\|^2. \tag{34}$$

For $0 \leq q \leq H$, let

$$R_q = \sum_{i=1}^{n}\sum_{s=0}^{q}\mathbb{E}\|\widetilde{h}_i^s(X_t^i)\|^2.$$

Observe that the inequality 33 can be rewritten as:

$$R_q - R_{q-1} \leq \frac{1}{2H}R_{q-1} + n(\sigma^2 + 4\rho^2) + 16L^2\mathbb{E}[\Gamma_t] + 4n\mathbb{E}\|\sum_{i=1}^{n}h_i^q(X_t^i)/n\|^2.$$

which is the same as

$$R_q \leq (1 + \frac{1}{2H})R_{q-1} + n(\sigma^2 + 4\rho^2) + 16L^2\mathbb{E}[\Gamma_t] + 4n\mathbb{E}\|\sum_{i=1}^{n}h_i^q(X_t^i)/n\|^2.$$

By unrolling the recursion we get that

$$R_H \leq \sum_{q=0}^{H-1}(1 + \frac{1}{2H})^q\Big(n(\sigma^2 + 4\rho^2) + 16L^2\mathbb{E}[\Gamma_t] + 4n\mathbb{E}\|\sum_{i=1}^{n}h_i^{H-q}(X_t^i)/n\|^2\Big)$$

Since,

$$(1 + \frac{1}{2H})^H \leq (e^{\frac{1}{2H}})^H = e^{1/2} \leq 2$$

we have that

$$R_H = \sum_{q=1}^{H}\sum_{i=1}^{n}\mathbb{E}\|\widetilde{h}_i^q(X_t^i)\|^2 \leq 2\Bigg(\sum_{q=1}^{H}\Big(n(\sigma^2 + 4\rho^2) + 16L^2\mathbb{E}[\Gamma_t] + 4n\mathbb{E}\|\sum_{i=1}^{n}h_i^q(X_t^i)/n\|^2\Big)\Bigg)$$

$$= 2Hn(\sigma^2 + 4\rho^2) + 32HL^2\mathbb{E}[\Gamma_t] + 8n\sum_{q=1}^{H}\mathbb{E}\|\sum_{i=1}^{n}h_i^q(X_t^i)/n\|^2.$$

$\square$

Next we derive the upper bound for $\sum_{t=0}^{T}\mathbb{E}[\Gamma_t]$:

**Lemma H.5.** *For $\eta \leq \frac{1}{10HL\sqrt{2r/\lambda_2 + 8r^2/\lambda_2^2}}$, we have that :*

$$\sum_{t=0}^{T}\mathbb{E}[\Gamma_t] \leq \frac{8nr\eta^2(\sigma^2 + 4\rho^2)H^2T}{\lambda_2}(2 + \frac{8r}{\lambda_2}) + \frac{32nr\eta^2H}{\lambda_2}(2 + \frac{8r}{\lambda_2})\sum_{t=1}^{T}\sum_{q=1}^{H}\mathbb{E}\|\sum_{i=1}^{n}h_i^q(X_t^i)/n\|^2.$$

*Proof.* By Lemma H.1 we get that:

$$\mathbb{E}[\Gamma_{t+1}] \le (1 - \frac{\lambda_2}{2rn})\mathbb{E}[\Gamma_t] + \frac{\eta^2 H}{n}(2 + \frac{8r}{\lambda_2})\sum_{q=1}^{H}\sum_{i=1}^{n}\mathbb{E}\|\widetilde{h}_i^q(X_t^i)\|^2$$

$$\overset{\text{Lemma H.4}}{\le} (1 - \frac{\lambda_2}{2rn})\mathbb{E}[\Gamma_t]$$

$$+ \frac{\eta^2 H}{n}(2 + \frac{8r}{\lambda_2})\Big(2Hn(\sigma^2 + 4\rho^2) + 32HL^2\mathbb{E}[\Gamma_t] + 8n\sum_{q=1}^{H}\mathbb{E}\|\sum_{i=1}^{n}h_i^q(X_t^i)/n\|^2\Big)$$

$$= (1 - \frac{\lambda_2}{2rn})\mathbb{E}[\Gamma_t] + 2\eta^2(\sigma^2 + 4\rho^2) + H^2(2 + \frac{8r}{\lambda_2}) + \frac{32H^2L^2\eta^2}{n}(2 + \frac{8r}{\lambda_2})\mathbb{E}[\Gamma_t]$$

$$+ 8\eta^2 H(2 + \frac{8r}{\lambda_2})\sum_{q=1}^{H}\mathbb{E}\|\sum_{i=1}^{n}h_i^q(X_t^i)/n\|^2$$

Notice that for $\eta \le \frac{1}{12HL\sqrt{2r/\lambda_2 + 8r^2/\lambda_2^2}}$ we can rewrite the above inequality as

$$\mathbb{E}[\Gamma_{t+1}] \le (1 - \frac{\lambda_2}{4nr})\mathbb{E}[\Gamma_t] + 2\eta^2(\sigma^2 + 4\rho^2)H^2(2 + \frac{8r}{\lambda_2}) + 8\eta^2 H(2 + \frac{8r}{\lambda_2})\sum_{q=1}^{H}\mathbb{E}\|\sum_{i=1}^{n}h_i^q(X_t^i)/n\|^2.$$

since $\sum_{i=0}^{\infty}(1 - \frac{\lambda_2}{4nr})^i \le \frac{1}{1 - (1 - \frac{\lambda_2}{4nr})} = \frac{4nr}{\lambda_2}$ we get that:

$$\sum_{t=0}^{T}\mathbb{E}[\Gamma_t] \le \frac{4nr}{\lambda_2}\Big(2(\eta^2 + 4\rho^2)\sigma^2 H^2(2 + \frac{8r}{\lambda_2}) + 8\eta^2 H(2 + \frac{4r}{\lambda_2})\sum_{q=1}^{H}\mathbb{E}\|\sum_{i=1}^{n}h_i^q(X_t^i)/n\|^2\Big)$$

$$= \frac{8nr\eta^2(\sigma^2 + 4\rho^2)H^2 T}{\lambda_2}(2 + \frac{8r}{\lambda_2}) + \frac{32nr\eta^2 H}{\lambda_2}(2 + \frac{8r}{\lambda_2})\sum_{t=1}^{T}\sum_{q=1}^{H}\mathbb{E}\|\sum_{i=1}^{n}h_i^q(X_t^i)/n\|^2.$$

$\square$

Now, we are ready to prove the following theorem:

**Theorem 4.2.** *Let $f$ be an non-convex, $L$-smooth function whose minimum $x^\star$ we are trying to find via the SwarmSGD procedure given in Algorithm 1. Let local functions of agents satisfy conditions (27), (28), (29), (30) and (31). Let $H$ be the number of local stochastic gradient steps performed by each agent upon interaction. Define $\mu_t = \sum_{i=1}^{n}X_t^i/n$, where $X_t^i$ is a value of model $i$ after $t$ interactions. For learning rate $\eta = \frac{n}{\sqrt{T}}$ and $T \ge 57600n^4H^2\max(1, L^2)(\frac{r^2}{\lambda_2^2} + 1)^2$ we have that:*

$$\frac{\sum_{i=0}^{T-1}\mathbb{E}\|\nabla f(\mu_t)\|^2}{T} \le \frac{1}{\sqrt{T}H}\mathbb{E}[f(\mu_0) - f(x^*)] + \frac{376H^2\max(1, L^2)(\sigma^2 + 4\rho^2)}{\sqrt{T}}(\frac{r^2}{\lambda_2^2} + 1).$$

*Proof.* Let $\mathbb{E}_t$ denote expectation conditioned on $\{X_1^t, X_2^t, ..., X_n^t\}$. By $L$-smoothness we have that

$$\mathbb{E}_t[f(\mu_{t+1})] \le f(\mu_t) + \mathbb{E}_t\langle\nabla f(\mu_t), \mu_{t+1} - \mu_t\rangle + \frac{L}{2}\mathbb{E}_t\|\mu_{t+1} - \mu_t\|^2. \tag{35}$$

First we look at $\mathbb{E}_t[\mu_{t+1} - \mu_t]$. If agents $i$ and $j$ interact (which happens with probability $\frac{1}{rn/2}$), we have that $\mu_{t+1} - \mu_t = -\frac{\eta}{n}\widetilde{h}_i(X_t^i) - \frac{\eta}{n}\widetilde{h}_j(X_t^j)$. Hence we get:

$$\mathbb{E}_t[\mu_{t+1} - \mu_t] = \frac{1}{rn/2}\sum_{(i,j)\in E}\mathbb{E}_t[-\frac{\eta}{n}\widetilde{h}_i(X_t^i) - \frac{\eta}{n}\widetilde{h}_j(X_t^j)] = \frac{2}{n}\sum_{i=1}^{n}\mathbb{E}_t[-\frac{\eta}{n}\widetilde{h}_i(X_t^i)]$$

$$= -\frac{2\eta}{n^2}\sum_{i=1}^{n}\mathbb{E}_t[\widetilde{h}_i(X_t^i)] \overset{(3)}{=} -\frac{2\eta}{n^2}\sum_{i=1}^{n}\sum_{q=1}^{H}\mathbb{E}_t[h_i^q(X_t^i)].$$

Using the above inequality we have:

$$\mathbb{E}_t \langle \nabla f(\mu_t), \mu_{t+1} - \mu_t \rangle = \langle \nabla f(\mu_t), \mathbb{E}_t[\mu_{t+1} - \mu_t] \rangle = \langle \nabla f(\mu_t), -\frac{2\eta}{n^2} \sum_{i=1}^{n} \sum_{q=1}^{H} \mathbb{E}_t[h_i^q(X_t^i)] \rangle$$

$$= \frac{\eta}{n} \sum_{q=1}^{H} \left( \mathbb{E}_t \|\nabla f(\mu_t) - \sum_{i=1}^{n} h_i^q(X_t^i)/n\|^2 - \|\nabla f(\mu_t)\|^2 - \mathbb{E}_t\|\sum_{i=1}^{n} h_i^q(X_t^i)/n\|^2 \right)$$

$$\overset{(28)}{=} \frac{\eta}{n} \sum_{q=1}^{H} \left( \mathbb{E}_t \|\sum_{i=1}^{n} (\nabla f_i(\mu_t) - h_i^q(X_t^i))/n\|^2 - \|\nabla f(\mu_t)\|^2 - \mathbb{E}_t\|\sum_{i=1}^{n} h_i^q(X_t^i)/n\|^2 \right)$$

$$\leq \frac{\eta}{n} \sum_{q=1}^{H} \left( \frac{1}{n} \sum_{i=1}^{n} \mathbb{E}_t \|\sum_{i=1}^{n} \nabla f_i(\mu_t) - h_i^q(X_t^i)\|^2 - \|\nabla f(\mu_t)\|^2 - \mathbb{E}_t\|\sum_{i=1}^{n} h_i^q(X_t^i)/n\|^2 \right).$$

Here we used Cauchy-Schwarz inequality at the last step. Next we look at $\mathbb{E}_t\|\mu_{t+1} - \mu_t\|^2$. If agents $i$ and $j$ interact, (which happens with probability $\frac{1}{rn/2}$). We have that $\mu_{t+1} - \mu_t = -\frac{\eta}{n}\widetilde{h}_i(X_t^i) - \frac{\eta}{n}\widetilde{h}_j(X_t^j)$. Hence we get that

$$\mathbb{E}_t\|\mu_{t+1} - \mu_t\|^2 = \frac{1}{rn/2} \sum_{(i,j)\in E} \mathbb{E}_t \left\| -\frac{\eta}{n}\widetilde{h}_i(X_t^i) - \frac{\eta}{n}\widetilde{h}_j(X_t^j) \right\|^2$$

$$\overset{Cauchy-Schwarz}{\leq} \frac{1}{rn/2} \sum_{(i,j)\in E} \frac{\eta^2}{n^2} \left( 2\mathbb{E}_t\|\widetilde{h}_i(X_t^i)\|^2 + 2\mathbb{E}_t\|\frac{\eta}{n}\widetilde{h}_j(X_t^j)\|^2 \right)$$

$$= \frac{2}{n} \sum_{i=1}^{n} \frac{2\eta^2}{n^2} \|\widetilde{h}_i(X_t^i)\| \overset{Cauchy-Schwarz}{\leq} \frac{4\eta^2 H}{n^3} \sum_{i=1}^{n} \sum_{q=1}^{H} \mathbb{E}_t\|\widetilde{h}_i^q(X_t^i)\|^2.$$

So, we can rewrite (35) as:

$$\mathbb{E}_t[f(\mu_{t+1})] \leq f(\mu_t) + \frac{\eta}{n} \sum_{q=1}^{H} \left( \frac{1}{n}\mathbb{E}_t\|\sum_{i=1}^{n} \nabla f(\mu_t) - h_i^q(X_t^i)\|^2 \right.$$

$$\left. - \|\nabla f(\mu_t)\|^2 - \mathbb{E}_t\|\sum_{i=1}^{n} h_i^q(X_t^i)/n\|^2 \right)$$

$$+ \frac{2L\eta^2 H}{n^3} \sum_{i=1}^{n} \sum_{q=1}^{H} \mathbb{E}_t\|\widetilde{h}_i^q(X_t^i)\|^2.$$

Next, we remove conditioning:

$$\mathbb{E}[f(\mu_{t+1})] = \mathbb{E}[\mathbb{E}_t[f(\mu_{t+1})]]$$

$$\leq \mathbb{E}[f(\mu_t)] + \frac{\eta}{n} \sum_{q=1}^{H} \Big( \frac{1}{n} \sum_{i=1}^{n} \mathbb{E}\|\nabla f_i(\mu_t) - h_i^q(X_t^i)\|^2$$

$$- \mathbb{E}\|\nabla f(\mu_t)\|^2 - \mathbb{E}\| \sum_{i=1}^{n} h_i^q(X_t^i)/n\|^2 \Big)$$

$$+ \frac{2L\eta^2 H}{n^3} \sum_{i=1}^{n} \sum_{q=1}^{H} \mathbb{E}\|\widetilde{h}_i^q(X_t^i)\|^2$$

$$\overset{\text{Lemma H.2}}{\leq} \mathbb{E}[f(\mu_t)] + \frac{\eta}{n} \sum_{q=1}^{H} \Big( \frac{1}{n}(2L^2\mathbb{E}[\Gamma_t] + \sum_{i=1}^{n} 2L^2\eta^2\mathbb{E}\| \sum_{s=0}^{q-1} \widetilde{h}_i^s(X_t^i)\|^2)$$

$$- \mathbb{E}\|\nabla f(\mu_t)\|^2 - \mathbb{E}\| \sum_{i=1}^{n} h_i^q(X_t^i)/n\|^2 \Big)$$

$$+ \frac{2L\eta^2 H}{n^3} \sum_{i=1}^{n} \sum_{q=1}^{H} \mathbb{E}\|\widetilde{h}_i^q(X_t^i)\|^2$$

$$\overset{\text{Lemma H.4}}{\leq} \mathbb{E}[f(\mu_t)] + \frac{2\eta L^2 H}{n^2}\mathbb{E}[\Gamma_t] - \frac{H\eta}{n}\mathbb{E}\|\nabla f(\mu_t)\|^2$$

$$+ \frac{2L^2\eta^3}{n^2}\Big( 2Hn(\sigma^2 + 4\rho^2) + 32HL^2\mathbb{E}[\Gamma_t] + 8n\sum_{q=1}^{H}\mathbb{E}\|\sum_{i=1}^{n}h_i^q(X_t^i)/n\|^2\Big)$$

$$+ \frac{2LH\eta^2}{n^3}\Big( 2Hn(\sigma^2 + 4\rho^2) + 32HL^2\mathbb{E}[\Gamma_t] + 8n\sum_{q=1}^{H}\mathbb{E}\|\sum_{i=1}^{n}h_i^q(X_t^i)/n\|^2\Big)$$

$$- \frac{\eta}{n}\sum_{q=1}^{H}\mathbb{E}\|\sum_{i=1}^{n}h_i^q(X_t^i)/n\|^2.$$

Next we choose $\eta \leq \frac{1}{9L}$ and $\eta \leq \frac{n}{80LH}$, so that $\frac{16L^2\eta^3}{n} \leq \frac{\eta}{5n}$ and $\frac{16LH\eta^2}{n^2} \leq \frac{\eta}{5n}$. This together with the above inequalities allows us to derive the following upper bound for $\mathbb{E}[f(\mu_{t+1})]$ (we eliminate terms with positive multiplicative factor $\mathbb{E}\|\sum_{i=1}^{n}h_i^q(X_t^i)/n\|^2$):

$$\mathbb{E}[f(\mu_{t+1})] \leq \mathbb{E}[f(\mu_t)] + \frac{2\eta L^2 H}{n^2}\mathbb{E}[\Gamma_t] - \frac{H\eta}{n}\mathbb{E}\|\nabla f(\mu_t)\|^2$$

$$+ \frac{2L^2\eta^3}{n^2}\Big( 2Hn(\sigma^2 + 4\rho^2) + 32HL^2\mathbb{E}[\Gamma_t]\Big)$$

$$+ \frac{2LH\eta^2}{n^3}\Big( 2Hn(\sigma^2 + 4\rho^2) + 32HL^2\mathbb{E}[\Gamma_t]\Big)$$

$$- \frac{3\eta}{5n}\sum_{q=1}^{H}\mathbb{E}\|\sum_{i=1}^{n}h_i^q(X_t^i)/n\|^2.$$

We proceed by summing up the above inequality for $0 \leq t \leq T-1$:

$$\sum_{t=0}^{T-1}\mathbb{E}[f(\mu_{t+1})] \leq \sum_{t=0}^{T-1}\mathbb{E}[f(\mu_t)] + \frac{4L^2 H\eta^3(\sigma^2 + 4\rho^2)T}{n} + \frac{4LH^2\eta^2(\sigma^2 + 4\rho^2)T}{n^2}$$

$$+ \frac{2\eta L^2 H}{n^2}\sum_{t=0}^{T-1}\mathbb{E}[\Gamma_t] + \frac{64L^4 H\eta^3}{n^2}\sum_{t=0}^{T-1}\mathbb{E}[\Gamma_t] + \frac{64L^3 H^2\eta^2}{n^3}\sum_{t=0}^{T-1}\mathbb{E}[\Gamma_t]$$

$$- \sum_{t=0}^{T-1}\frac{3\eta}{5n}\sum_{q=1}^{H}\mathbb{E}\|\sum_{i=1}^{n}h_i^q(X_t^i)/n\|^2 - \sum_{i=0}^{T-1}\frac{\eta H}{n}\mathbb{E}\|\nabla f(\mu_t)\|^2. \tag{36}$$

Next we use Lemma H.5:

$$
\frac{2\eta L^2 H}{n^2} \sum_{t=0}^{T-1} \mathbb{E}[\Gamma_t] + \frac{64L^4 H \eta^3}{n^2} \sum_{t=0}^{T-1} \mathbb{E}[\Gamma_t] + \frac{64L^3 H^2 \eta^2}{n^3} \sum_{t=0}^{T-1} \mathbb{E}[\Gamma_t]
$$

$$
\leq \frac{2\eta L^2 H}{n^2} \left( \frac{8nr\eta^2(\sigma^2 + 4\rho^2)H^2 T}{\lambda_2}(2 + \frac{8r}{\lambda_2}) \right.
$$

$$
\left. + \frac{32nr\eta^2 H}{\lambda_2}(2 + \frac{8r}{\lambda_2}) \sum_{t=1}^{T-1} \sum_{q=1}^{H} \mathbb{E}\| \sum_{i=1}^{n} h_i^q(X_t^i)/n\|^2 \right)
$$

$$
+ \frac{64L^4 H \eta^3}{n^2} \left( \frac{8nr\eta^2(\sigma^2 + 4\rho^2) + H^2 T}{\lambda_2}(2 + \frac{8r}{\lambda_2}) \right.
$$

$$
\left. + \frac{32nr\eta^2 H}{\lambda_2}(2 + \frac{8r}{\lambda_2}) \sum_{t=1}^{T-1} \sum_{q=1}^{H} \mathbb{E}\| \sum_{i=1}^{n} h_i^q(X_t^i)/n\|^2 \right)
$$

$$
+ \frac{64L^3 H^2 \eta^2}{n^3} \left( \frac{8nr\eta^2(\sigma^2 + 4\rho^2) + H^2 T}{\lambda_2}(2 + \frac{8r}{\lambda_2}) \right.
$$

$$
\left. + \frac{32nr\eta^2 H}{\lambda_2}(2 + \frac{8r}{\lambda_2}) \sum_{t=1}^{T-1} \sum_{q=1}^{H} \mathbb{E}\| \sum_{i=1}^{n} h_i^q(X_t^i)/n\|^2 \right).
$$

By choosing $\eta \leq \frac{1}{18HL\sqrt{2r/\lambda_2 + 8r^2/\lambda_2^2}}$, $\eta \leq \frac{1}{11H^{1/2}L(2r/\lambda_2 + 8r^2/\lambda_2^2)^{1/4}}$ and $\eta \leq \frac{n^{1/3}}{22LH(2r/\lambda_2 + 4r^2/\lambda_2^2)^{1/3}}$ we can eliminate terms with the multiplicative factor $\sum_{q=1}^{H} \mathbb{E}\| \sum_{i=1}^{n} h_i^q(X_t^i)/n\|^2$ in the inequality (36):

$$
\sum_{t=0}^{T-1} \mathbb{E}[f(\mu_{t+1})] \leq \sum_{t=0}^{T-1} \mathbb{E}[f(\mu_t)] - \sum_{i=0}^{T-1} \frac{\eta H}{n} \mathbb{E}\|\nabla f(\mu_t)\|^2
$$

$$
+ \frac{4L^2 H \eta^3 (\sigma^2 + 4\rho^2) T}{n} + \frac{4LH^2 \eta^2 (\sigma^2 + 4\rho^2) T}{n^2} +
$$

$$
+ \frac{16\eta^3 L^2 H^3 T (\sigma^2 + 4\rho^2)}{n}(2r/\lambda_2 + 8r^2/\lambda_2^2)
$$

$$
+ \frac{512L^4 H^3 \eta^5 T (\sigma^2 + 4\rho^2)}{n}(2r/\lambda_2 + 8r^2/\lambda_2^2)
$$

$$
+ \frac{512L^3 H^4 \eta^4 (\sigma^2 + 4\rho^2) T}{n^2}(2r/\lambda_2 + 8r^2/\lambda_2^2).
$$

After rearranging terms and dividing by $\frac{\eta T H}{n}$ we get that

$$
\frac{\sum_{i=0}^{T-1} \mathbb{E}\|\nabla f(\mu_t)\|^2}{T} \leq \frac{n}{\eta T H} \mathbb{E}[f(\mu_0) - f(\mu_t)] + 4L^2 \eta^2 (\sigma^2 + 4\rho^2) + \frac{4LH\eta(\sigma^2 + 4\rho^2)}{n}
$$

$$
+ 16\eta^2 L^2 H^2 (\sigma^2 + 4\rho^2)(2r/\lambda_2 + 8r^2/\lambda_2^2)
$$

$$
+ 512L^4 H^2 \eta^4 (\sigma^2 + 4\rho^2)(2r/\lambda_2 + 8r^2/\lambda_2^2)
$$

$$
+ \frac{512^3 H^3 \eta^3 (\sigma^2 + 4\rho^2)}{n}(2r/\lambda_2 + 8r^2/\lambda_2^2).
$$

Next we use $\eta \leq 1/n$ and $\eta \leq \frac{1}{6HL}$:

$$
\begin{aligned}
\frac{\sum_{i=0}^{T-1} \mathbb{E}\|\nabla f(\mu_t)\|^2}{T} &\leq \frac{n}{\eta TH}\mathbb{E}[f(\mu_0) - f(\mu_t)] + \frac{4L^2\eta(\sigma^2 + 4\rho^2)}{n} + \frac{4LH\eta(\sigma^2 + 4\rho^2)}{n} \\
&\quad + \frac{16\eta L^2 H^2(\sigma^2 + 4\rho^2)}{n}(2r/\lambda_2 + 4r^2/\lambda_2^2) \\
&\quad + \frac{384L^4 H^2 \eta^3(\sigma^2 + 4\rho^2)}{n}(2r/\lambda_2 + 8r^2/\lambda_2^2) \\
&\quad + \frac{384L^3 H^3 \eta^3 \sigma^2}{n}(2r/\lambda_2 + 8r^2/\lambda_2^2) \\
&\leq \frac{n}{\eta TH}\mathbb{E}[f(\mu_0) - f(\mu_t)] + \frac{4L^2\eta(\sigma^2 + 4\rho^2)}{n} + \frac{4LH\eta(\sigma^2 + 4\rho^2)}{n} \\
&\quad + \frac{16\eta L^2 H^2(\sigma^2 + 4\rho^2)}{n}(2r/\lambda_2 + 8r^2/\lambda_2^2) \\
&\quad + \frac{15L^2\eta(\sigma^2 + 4\rho^2)}{n}(2r/\lambda_2 + 8r^2/\lambda_2^2) \\
&\quad + \frac{15LH\eta\sigma^2}{n}(2r/\lambda_2 + 8r^2/\lambda_2^2)
\end{aligned}
$$

.

Recall that $\eta = \frac{n}{\sqrt{T}}$ to get:

$$
\begin{aligned}
\frac{\sum_{i=0}^{T-1} \mathbb{E}\|\nabla f(\mu_t)\|^2}{T} &\leq \frac{1}{\sqrt{T}H}\mathbb{E}[f(\mu_0) - f(\mu_t)] + \frac{4L^2(\sigma^2 + 4\rho^2)}{\sqrt{T}} + \frac{4LH(\sigma^2 + 4\rho^2)}{\sqrt{T}} \\
&\quad + \frac{16L^2 H^2(\sigma^2 + 4\rho^2)}{\sqrt{T}}(2r/\lambda_2 + 8r^2/\lambda_2^2) \\
&\quad + \frac{15L^2(\sigma^2 + 4\rho^2)}{\sqrt{T}}(2r/\lambda_2 + 8r^2/\lambda_2^2) + \frac{15LH\sigma^2}{\sqrt{T}}(2r/\lambda_2 + 8r^2/\lambda_2^2) \\
&\leq \frac{1}{\sqrt{T}H}\mathbb{E}[f(\mu_0) - f(\mu_t)] + \frac{376H^2 \max(1, L^2)(\sigma^2 + 4\rho^2)}{\sqrt{T}}(\frac{r^2}{\lambda_2^2} + 1) \\
&\leq \frac{1}{\sqrt{T}H}\mathbb{E}[f(\mu_0) - f(x^*)] + \frac{376H^2 \max(1, L^2)(\sigma^2 + 4\rho^2)}{\sqrt{T}}(\frac{r^2}{\lambda_2^2} + 1).
\end{aligned}
$$

where in the last step we used $f(\mu_t) \geq f(x^*)$. Notice that all assumptions and upper bounds on $\eta$ are satisfied if

$$
\eta \leq \frac{1}{240nH \max(1, L)(\frac{r^2}{\lambda_2^2} + 1)}, \tag{37}
$$

which is true

$$
T \geq 57600n^4 H^2 \max(1, L^2)(\frac{r^2}{\lambda_2^2} + 1)^2. \tag{38}
$$

$\square$

# I ADDITIONAL EXPERIMENTAL RESULTS

We validated our analysis, by applying the algorithm to training deep neural networks for image classification and machine translation.

**Target System and Implementation.** We run SwarmSGD on the CSCS Piz Daint supercomputer, which is composed of Cray XC50 nodes, each with a Xeon E5-2690v3 CPU and an NVIDIA Tesla P100 GPU, using a state-of-the-art Aries interconnect. Please see (Piz, 2019) for hardware details.

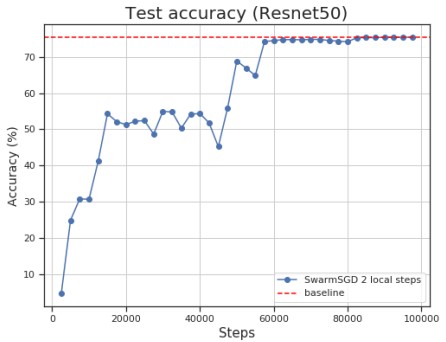 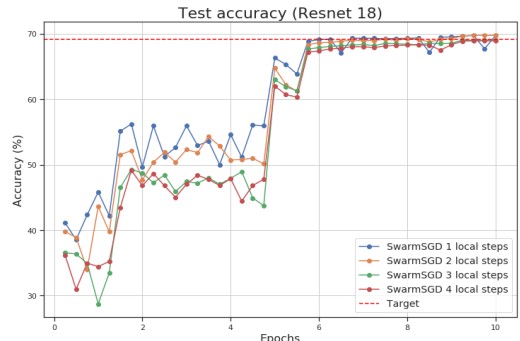

(a) Convergence of ResNet50/ImageNet versus number of gradient steps. SwarmSGD is able to recover the baseline top accuracy.

(b) Convergence versus number of local steps for ResNet18 on ImageNet. All variants recover the target accuracy, but we note the lower convergence of variants with more local steps.

Figure 3: Additional convergence results for ImageNet dataset.

We implemented SwarmSGD in Pytorch and TensorFlow using NCCL/MPI respectively. Basically, each node implements a computation thread, and a communication thread, each of which stores a copy of the model. The "live" copy, which is being updated with gradients, is stored by the computation thread. Periodically, the threads synchronize their two models. When interacting, the two nodes exchange model information via their communication threads. Our implementation closely follows the non-blocking Swarm algorithm description.

We used SwarmSGD to train ResNets on the classic CIFAR-10/ImageNet datasets, and a Transformer Vaswani et al. (2017) on the WMT17 dataset (English-Germa). The code will be made available upon publication.

**Hyperparameters.** The only additional hyperparameter is the total number of epochs we execute for. Once we have fixed the number of epochs, *we do not alter the other training hyperparameters*: in particular, the learning rate schedule, momentum and weight decay terms are identical to sequential SGD, for each individual model. Practically, if sequential SGD trains ResNet18 in 90 epochs, decreasing the learning rate at 30 and 60 epochs, then SwarmSGD with 32 nodes and multiplier 2 would $90 * 2/32 \simeq 5.6$ epochs per node, decreasing the learning rate at 2 and 4 epochs.

Specifically, for the ImageNet experiments, we used the following hyper-parameters. For ResNet18 and ResNet50, we ran for $240$ total parallel epochs using 32 parallel nodes. The first communicated every 3 local steps, whereas the second communicated every 2 local steps. We used the same hyperparameters (initial learning rate 0.1, annealed at 1/3 and 2/3 through training, and standard weight-decay and momentum parameters).

For the WMT17 experiments, we ran a standard Transformer-large model, and executed for 10 *global* epochs at 16, 32, and 64 nodes. We ran a version with multiplier 1 (i.e. $10/\text{NUM\_NODES}$ epochs per model) and one with multiplier 1.5 (i.e. $15/\text{NUM\_NODES}$ epochs per model) and registered the BLEU score for each.

**Baselines.** We consider the following baselines:

- **Data-parallel SGD:** Here, we consider both the small-batch (strong scaling) version, which executes a global batch size of 256 on ImageNet/CIFAR experiments, and the large-batch (weak-scaling) baseline, which maximizes the batch per GPU. For the latter version, the learning rate is tuned following Goyal et al. (2017).

- **Local SGD:** Stich (2018); Lin et al. (2018) We follow the implementation of Lin et al. (2018), communicating globally every 5 SGD steps (which was the highest setting which provided good accuracy on the WMT task).

- **Previous decentralized proposals:** We experimented also with D-PSGD Lian et al. (2017), AD-PSGD Lian et al. (2018), and SGP Assran et al. (2018). Due to computational constraints, we did not always measure their end-to-end accuracy. Our method matches the sequential / large-batch accuracy for the models we consider within $1\%$. We note that the

best performing alternative (AD-PSGD) is known to drop accuracy relative to the baselines, e.g. (Assran et al., 2018).

**Results.** The accuracy results for ImageNet experiments are given in Table 1 and Figures 3(a) and 3(b). As is standard, we follow Top-1 validation accuracy versus number of steps.

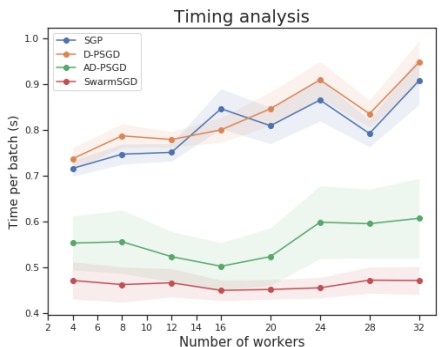

Figure 4: Average time per batch for previous methods, compared to SwarmSGD, on ResNet18/ImageNet, across 1000 repetitions with warm-up. Notice that 1) the time per batch of SwarmSGD stays *constant* relative to the number of nodes; 2) it is lower than any other method. This is due to the *reduced communication frequency*. Importantly, the base value on the y axis of this graph (0.4) is the average computation time per batch. Thus, everything above 0.4 represents the average communication time for this model.

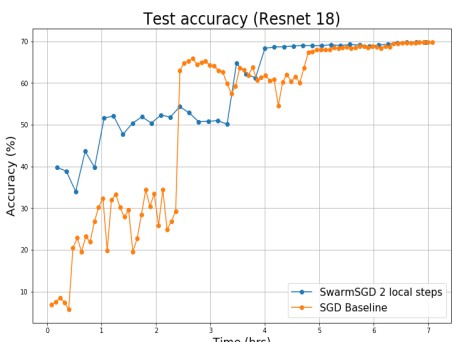

Figure 5: Convergence versus time for ResNet18/Imagenet for the SGD baseline vs Swarm, executing at 32 nodes. We note that Swarm iterates for $2.7\times$ more epochs for convergence, which explains the similar runtime despite the better scalability of Swarm.

**Communication cost.** We now look deeper into SwarmSGD's performance. For this, we examine in Figure 4 the average time per batch of different methods when executed on our testbed. The base value on the y axis (0.4s) is exactly the *average time per batch*, which is the same across all methods. Thus, the extra values on the y axis equate roughly to the communication cost of each algorithm. The results suggest that the communication cost can be up to twice the batch cost (SGP and D-PSGD). Moreover, this cost is *increasing* when considered relative to the number of workers (X axis), for all methods except SwarmSGD.

This reduced cost is justified simply because our method *reduces communication frequency:* it communicates less often, and therefore the average cost of communication at a step is lower. We can therefore conclude that our method is *scalable*, in the sense that its communication cost remains constant relative to the total size of the system. Figure 3(b) shows the convergence versus time for ResNet18 on the ImageNet dataset, at 32 nodes, with 3 local steps per node, and $\sim 7$ epochs per model.

**Convergence versus Steps and Epochs.** Figure 8 shows and discusses the results of additional ablation studies with respect to the number of nodes/processes and number of local steps / total epochs on the CIFAR-10 dataset / ResNet20 model. In brief, the results show that the method still

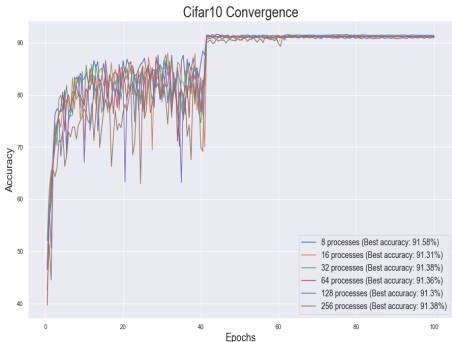
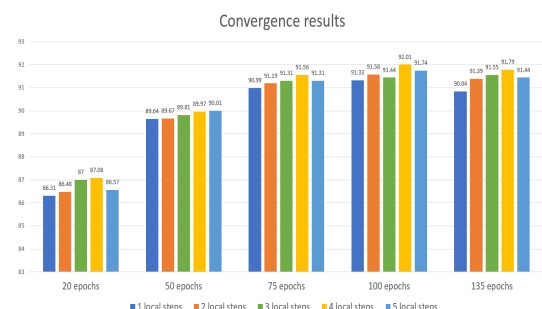

(a) Convergence versus number epochs (per model) for CIFAR-10/ResNet20, at node counts between 8 and 256. We note that the algorithm converges and recovers SGD accuracy (91.35% Top-1) for all node counts, although there are oscillations at high node counts.

(b) Accuracy versus local epochs and local steps for CIFAR-10/ResNet20. The original schedule for this model has 300 epochs, and this experiment is executed on 8 nodes. If the convergence scaling were perfect, $300/8 = 37.5$ epochs would have been sufficient to converge. However, in this case we need an epoch multiplier of 2, leading to 75 epochs to recover full accuracy (which in this case is 91.35%).

Figure 6: Additional convergence results for CIFAR-10 dataset, versus number of nodes (left), and local steps (right).

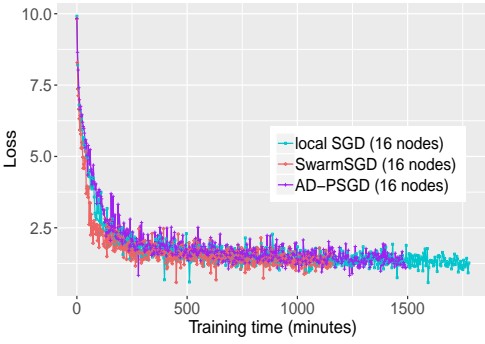

Figure 7: Objective loss versus time for the Transformer-XL/WMT experiment, for various methods, executing at 16 nodes.

preserves convergence even at very high node counts (256), and suggest a strong correlation between accuracy and the number of epochs executed per model. The number of local steps executed also impacts accuracy, but to a much lesser degree.

**Quantization.** Finally, we show convergence and speedup for a WideResNet-28 model with width factor 2, trained on the CIFAR-10 dataset. We note that the epoch multiplier factor in this setup is 1, i.e. Swarm (and its quantized variant) execute exactly the same number of epochs as the baseline. Notice that the quantized variant provides approximately 10% speedup in this case, for a $< 0.3\%$ drop in Top-1 accuracy.

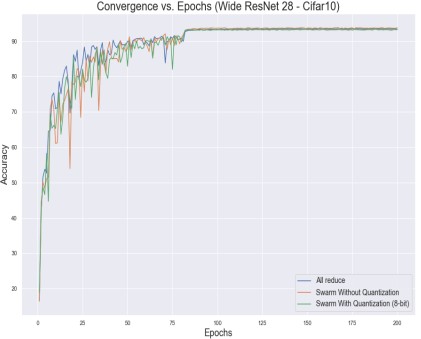

(a) Convergence versus number of steps for the quantized variant.

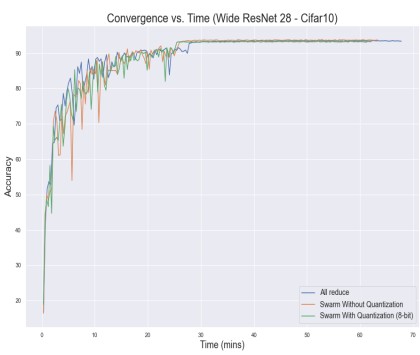

(b) Convergence versus time .

Figure 8: Convergence results for quantized 2xResNet28 trained on the CIFAR-10 dataset, versus iterations (left), and time (right).

