# OpenReview forum: "Decentralized SGD with Asynchronous, Local and Quantized Updates"
_ICLR.cc/2021/Conference — Reject_

### Official Review · AnonReviewer2 · 2020-10-25

**Rating:** 5
**Confidence:** 5

**Review:**

Summary: This paper combines the existing scaling techniques to reduce the communication cost of distributed SGD among a large number of computing nodes. These techniques include asynchronous, decentralized, or quantized communication. The authors prove that this combined algorithm converges to a local optimal point. In the experiments, this algorithm also successfully converges and scales for big data. The authors claim that this is the first work to consider decentralization, local updates, asynchrony, and quantization in conjunction.

Overall, the contribution of this paper is relatively marginal. The algorithm simply combines many different existing techniques and does not lead to any substantial new development. Below are some comments and questions.

(1) The first two paragraphs of the introduction look wordy. They introduced the distributed SGD problem and listed the scaling techniques as well as the relevant literature, but the meaning of these techniques is unclear. How these techniques are applied and combined is also unclear.

(2) The meaning of n and T are not formally defined.

(3) In Theorem 4.1, the assumption that T>=n^4 (n^4 can be very large) is the disadvantage of this algorithm because the same convergence rate O(1/sqrt(T)) has been achieved without such assumption in some distributed settings, including plain distributed SGD, federated average, etc.

(4) The claim in the abstract that the new algorithm can converge to local minima is not supported, since the theorems only imply gradient convergence.

(5) In the theoretical part, I did not see in which measure does this new algorithm excel the existing ones. The authors should clarify this. In the experiments, the objective function value of interest is not compared.

(6) On page 2, the authors said “SwarmSGD has a Θ(n) speedup in the non-convex case, matching results from previous work which considered decentralized dynamics but which synchronize upon every SGD step.” What is the measure, is it the number of communications, local SGD iterations or gradient evaluations? “Matching results” can be interpreted as equal to the previous rate, which seems to contradict with Θ(n) speedup. Please clarify this.

(7) In the contribution part, the authors mention that their new algorithm has lower average synchronization cost per iteration but more iterations in the experiments, how about the total synchronization cost?

(8) The authors use multiple variables to denote the number of nodes, including n, P and m. Please use only one.

(9) The space around the section captions is too narrow. This is not suggested in general.

---

> ### Author Response · Authors · 2020-11-14
> **Individual response**
>
> Thank you for your feedback. We address the main issues below, in order:
>
> > 0. “The algorithm simply combines many different existing techniques and does not lead to any substantial new development.”
>
> We agree that our algorithm is a combination of previous techniques.
> However, we would like to mention the following points:
>
> ●	First, our main theoretical contribution is on the analysis side: our technique is the first to be able analyze all four consistency relaxations (decentralization, asynchrony, quantization, and local steps) in conjunction, using a single type of argument. The fact that this a significant challenge is recognized by the community: please see, for instance, [“Advances and Open Problems in Federated Learning” arXiv:1912.04977], Section 2.1.2, which lists these possible consistency relaxations and poses their joint analysis as a challenge.
>
> ●	Second, we would argue that combining these existing techniques is not always trivial. One particularly tricky example is adding quantization, which is known to be challenging even in the basic decentralized setting (without asynchrony).
>
> ●	Third, results suggest that our method outperforms previous decentralized proposals in terms of accuracy-versus-time on practically-relevant models.
>
> Thank you for the detailed comments on the presentation, which we have addressed as follows. (We follow your numbering.)
>
> *1.	We have compressed the introductory paragraphs and put them in context, as well as substantially revised the introduction for clarity.*
>
> *2.	We have defined n and T upfront formally.*
>
> 3.	You are right that centralized or synchronous settings do not require $T \ge n^4$.
> However, as discussed in the answer to AnonReviewer 3, first point, a non-trivial bound on T is required for mixing in the *decentralized* setting and our requirement is the least strict among existing methods (e.g. AD-PSGD requires $T\ge n^6$, and SGP requires $T \ge n d^2$). Please see Appendix B for a detailed discussion.
>
> *We have added a discussion on this in the body, and invite the reviewer to examine Appendix B for a detailed discussion of the assumptions and of the relation to prior work.*
>
> *4.	Fixed.*
>
> 5.	Theoretically, our algorithm should dominate previous decentralized proposals in terms of total communication steps to convergence and total communication cost. Practically, it dominates them in terms of time-to-accuracy (see Figure 1(a)).
>
> *We have added a graph of loss-vs-time for the Transformer example. Since the trends were identical to the BLEU graph, this is given in the Appendix.*
>
> 6.	Good point. The measure is the total number of communication steps, but this is equivalent up to constants to the total number of gradient evaluations and to the total number of iterations, as we usually assume H to be a constant.
>
> *We have added a discussion of the convergence trade-off induced by $H$.*
>
> *7.	Also a good point.
> In terms of the theoretical analysis, if we disregard quantization and local steps ($H = 1$), our bounds are equivalent to those of [Lian et al., 2017].*
>
> *In practice, our algorithm is superior to previous proposals in terms of total communication cost, everything else being equal, especially at high node counts. For an illustration, please see Figure 2(b). Our algorithm has similar or better accuracy for the same number of gradient evaluations relative to previous proposals, and its cost per communication step is significantly lower than SGP or even AD-PSGD. The same point is made in Figure 1(b)--please see the results at 64 nodes.*
>
> *8.	and 9.: We apologize for these issues, which we have fixed in the revision.*

---

### Official Review · AnonReviewer1 · 2020-10-28
**several unclear places; strong assumption on the graph**

**Rating:** 6
**Confidence:** 4

**Review:**

This paper considers several techniques to minimize decentralized cost for training neural network models via stochastic gradient descent (SGD). These techniques include asynchronization, local updates, and quantized communication. Theoretical convergence analyses are provided, and numerical experiments are shown.

Strength:

- The provided convergence rate is well-separated and well-explained, though the reviewer did not check the correctness of all the proofs.
- Combining these techniques into decentralized SGD is new to the best of the reviewer's knowledge.

Weakness:

- The number of graphs satisfying the property is very limited. It requires an r-regular graph. That is, the number of edges connected to one node is the same for all nodes. This condition is very difficult to satisfy in applications. Therefore, the application would be limited too.
- The quantization part is limited comparing to the other two parts. What does the effect of quantization on the convergence rate and the communication cost? What is the benefit of using the quantization method in Davies et al. (2020)?
- In the proposed algorithm, each time an edge is activated and the two nodes connected through the edge are updated. Therefore, there is still synchronization in Alg. 1. Whether is it possible to update one node based on the results from multiple connected nodes (i.e., one node is activated)?
- Algorithm 2 is unclear. 'avg' is computed but not used. What are j' and 'i''?


## Update

The authors' response addresses some concerns, and I would like to keep the initial scores.

---

> ### Author Response · Authors · 2020-11-14
> **Individual response**
>
> [*We mark updates to the response in italics.*]
>
> Thank you for your feedback. We address the main issues below, in order:
>
> >1.	“An $r$-regular graph is required.”
>
> This assumption faithfully models supercomputing and cloud networks, which tend to be densely connected and low-diameter.
> For example, Dragonfly topologies are very popular in supercomputing networks, and are regular.
>
> *We have added references motivating this modeling choice in the revision. More generally, we believe our technique can be used to analyze the process on general graphs, which we leave for future work.*
>
> >2.	The benefit of using the quantization scheme of [Davies et al.]
>
> **Short answer:**
>
> We use [Davies et al.] because it is the only quantization scheme where the error depends on the *distance* between its inputs, and not on the *norm* of its inputs. This is a critical issue when quantizing models, since we do not have a bound on their norms (as opposed to quantizing gradients, for which we could use the second-moment bound), but we do have a bound on their difference (via Gamma).
> Even so, we have to be very careful in the parametrization of this quantization scheme to achieve convergence.
>
> *We have added a specific discussion on this point in the revision.*
>
> **Longer answer:**
>
> [Davies et al] allows us to bound the error caused by quantization by $\|X_t^i-X_t^j\|^2$ (if we quantize $X_t^i$ and send it to $j$), which in turn can be bounded naturally via our bound on $\Gamma(t)$. (Please see Appendix G, analysis outline paragraph for a more detailed overview.)
> Standard quantization schemes, such as QSGD, bound the error by $\|X_t^i\|^2$. Using such a scheme, it will be difficult to show the convergence, since $X_t^i$ is the model and we do not have any guarantees on its absolute norm: we just know that models are not far from each other, but they could be arbitrarily large, which would lead to arbitrary error.
>
> > “The effect of quantization on convergence and communication cost.”
>
>
> *We have added a discussion about the communication cost and the effect of quantization. In short, the convergence bounds stay exactly the same as in Theorem 4.1. (See Theorem G.1 in the Appendix).*
>
> > 3.	“Algorithm 2 is not clear.“
>
> We apologise for the lack of clarity, avg is indeed not needed in this case. $i$ and $j$ are the nodes which interact at step $t$. $X^i-S^i$ is the local steps node $i$ performed (initially the model had value $X^i$ and after local steps value is $S^i$). $X^i$ gets averaged with an estimate of model $X^j$ (denoted by ${X^j}’$) and only after that we apply local steps.
>
> > 4.	“Each time an edge is activated and the two nodes connected through the edge are updated. Therefore, there is still synchronization in Alg. 1.”
>
> Exactly! This is precisely the limitation we remove in Extension 2 (Non-blocking averaging), which allows a method to just “push” its model update to its communication partner in a non-blocking way, and to move on to the next iteration.
>
> > “Is it possible to update one node based on the results from multiple connected nodes?”
>
> This is possible, but the node would have to complete its local steps corresponding to its first interaction before the second interaction partner may communicate with it.

---

### Official Review · AnonReviewer3 · 2020-10-28
**Proof issues.**

**Rating:** 5
**Confidence:** 4

**Review:**

# Contributions:
1. This paper analyzes the convergence of decentralized SGD with asynchronous updates, quantization and local updates, which is novel and challenging.

2. The proposed algorithm requires significantly less communications to converge.

3. The authors have done extensive analysis of the convergence under different settings with detailed proofs.

4. The authors have done some large-scale experiments and show their algorithm performs great in practice.


# Strong points:

1. The authors have done concrete non-trivial analysis.

2. The algorithm is very general, several existing algorithms can be its special cases by different choice of parameters.

3. The experiment section provides a large amount of empirical evidence.


# Weak points:

1. Assumptions are too strong for Theorem 4.1 and 4.2:

	- Assuming each node can sample from global data is too strong. Section I removes this assumption but without highlighting key steps.

	- Step size requires the knowledge of the number of total steps.

	- Number of total steps needs to be larger than $n^4$. Even nodes don't communicate, the algorithm should still converge because the global sampling.

2. The benefit of local steps is not clear. For example, if we optimize the convergence rate in Theorem 4.1 over $H$, the best choice is $H = \Big(\frac{\lambda_2^2}{r^2} \cdot \frac{f(\mu_0) - f^*}{L^2 M^2} \Big)^{1/3}$. That is, the optimal $H$ is smaller when $r$ is larger.

3. The $H^2$ term in Theorem 4.1 and 4.2 may not be good enough. If set $H \to \infty$, then this bound should reduce to the single-machine SGD. However, the $H^2$ term will go to $\infty$.
4. Theorem 4.2 requires $T \sim O(*)$. Does it work if $T$ is greater?


5. Definition of $T$ is confusing.

6. Arguments for acceleration is not convincing. The algorithm only have one pair of nodes communicate, it's not clear how to replace $T$ with $nT$.


# Recommendation:

Weak reject. As of the current version, the proofs need to be improved. However, I believe the authors can improve in the next version.



# Further questions:

1. Is it possible to merge Section I with Theorem 4.1 or show the proof? I think there will be one term that depends on $\rho^2$. When $\rho^2 = 0$, Section I will reduce to Theorem 4.1.

2. Lemma F.3 is confusing. I think $\Gamma_t$ should decrease with $t$, or use diminishing step size $\eta_t$ to control this term. Then there's no need to set $\eta \sim \frac{1}{\sqrt{T}}$.

3. Can you also show the run time plot for ResNet?


# Optional improvements:

 1. It may be better to remove some small terms to make rate more clearer. For example,
	 - For Theorem 4.1, use $1 \leq \frac{r^2}{\lambda_2^2}$ can get rid of the constant $1$.
	 - For (14) and (19), use $\frac{r}{\lambda_2} \leq \frac{r^2}{\lambda_2^2}$ to get rid of the first order term.

2. The 3rd equation in Section D, $\tilde h_i^s$ also depends on $\tilde g_i$, which is not reflected.

3. The 1st equation in Section E has an extra '-'.

4. Is the coefficient $\frac{n - 2}{n}# in Eq (18) missing?

# Update

Thanks for the authors to address my questions. However, if the analysis can not explain why more local updates can reduce communications, I would not recommend to accept.

---

> ### Author Response · Authors · 2020-11-14
> **Individual response**
>
> [*The response was updated to reflect the contents of the revision. The modifications are described in italics.*]
>
> Thank you for your feedback. We address the main issues below, in order:
>
> A.	Weak points.
>
> 1.	“The $T \geq n^4$ bound on the total number of steps.”
>
> Please note that, intuitively, a non-trivial bound on T is necessary in the decentralized case since the averaging process has to “mix” in order to transfer model information. As such, all previous algorithms require such a bound.
> Specifically, in the case of [Lian et al, 2017, AD-PSGD], this bound is $T \geq n^6$, whereas [Assran et al, SGP]  requires $T \ge n d^2$, where $d$ is the *dimension* parameter, which is much larger than n in our applications. From this point of view, our conditions are the least restrictive, and they do hold in the practical setup we consider (For our ResNet/ImageNet experiments, T \ge 170K, and n^4 = 65K.)
>
> *We have added a discussion specifically on this point as part of our main theorem.
> For a very detailed discussion, please also see Appendix section B, which presents a detailed comparison with prior work in terms of assumptions.*
>
> 2.	“Step size requires the knowledge of the number of total steps.”
>
> This is a common assumption in this setting, see e.g. [D-PSGD, AD-PSGD, SGP]. Additionally, nodes could simply fix $T$ so that they know the learning rate, and run the algorithm so that the total number of interactions is at least $T$, but at most $2T$ . This, for example, can be done by stopping the algorithm after the first time *some* node reaches $2T/n$ interactions. A simple probabilistic argument will ensure that the mentioned bounds hold for the *total number of interactions*. This modification would only change the constants in the final convergence bounds.
>
> 3.	“Sampling from global data.”
>
> As the reviewer noticed, we did provide an analysis without this assumption in the Appendix.
> To avoid handwaving we rewrote Theorem 4.2 with non-i.i.d data distribution in mind.
> The main difference, which changes convergence bounds, is in the proof of Lemma H.3.
> The key change is following.
> With i.i.d data we are bounding $\sum_{i=1}^n \| \nabla f(\mu_t)-\sum_{j=1}^n \nabla f(X_t^j)/n\|^2$
> by $n \| \nabla f(\mu_t)-\sum_{j=1}^n \nabla f(X_t^j)/n\|^2 = 1/n \|\sum_{j=1}^n (\nabla f(\mu_t)-\nabla f(X_t^j)) \|^2 \le L^2 \Gamma_t$ where we used Cauchy-Schwarz and L-smoothness of f.
> with non-i.i.d data we need to bound $\sum_{i=1}^n \|\nabla f_i(\mu_t)-\sum_{j=1}^n \nabla f_j(X_t^j)/n\|^2$.
> We use that it is equal to $\sum_{i=1}^n \| \nabla f_i(\mu_t)-\nabla f(\mu_t)+\sum_{j=1}^n \nabla f_j (\mu_t)/n-\sum_{j=1}^n \nabla f_j(X_t^j)/n\|^2$. After using Cauchy-Schwarz $\sum_{i=1}^n \|\nabla f_i(\mu_t)-\nabla f(\mu_t)\|^2$ can be bounded by
> variance term $\rho^2n$ and $ \sum_{i=1}^n \|\sum_{j=1}^n \nabla f_j (\mu_t)/n-\sum_{j=1}^n \nabla f_j(X_t^j)/n \|^2$ can be
> bounded as in the case of i.i.d data since local functions $f_i$ are L-smooth as well.
>
> 4.	“The Benefit of local steps is not clear.”
>
> *One key benefit of local steps is practical, since it reduces average communication cost. However, there is also a theoretical benefit:
> the first term in the bound of Theorem 4.1 gets divided by $H$, which means that the algorithm does take advantage of all local steps. At the same time, local steps do increase the second "variance" term in the bound. Thus, the exact benefit-versus-costs analysis will depend on the exact problem parameters. We have added a detailed discussion of this point after the statement of Theorem 4.1.*

---

> > ### Author Response · Authors · 2020-11-14
> > **Individual response**
> >
> > A.	Weak points (continued).
> >
> >
> > 5.	“$H^2$ term in the convergence bound.”
> >
> > This is a good point. The $H^2$ term usually comes from using Cauchy-Schwarz in order to bound second moment of sum of $H$ gradients (so that we upper bound the potential Gamma, which measures the disbalance between local models), which would not be needed if $H$ goes to infinity and nodes never communicate. However, in this case it is not clear that it is possible to provide any guarantees on the convergence of the mean $\mu_t$ of the models  in the non-convex case.
> >
> > 6.	“Theorem 4.2 requires, $T=O(*)$.”
> >
> > Thank you for pointing this out, $O$ should be replaced with $\ge$ and the Theorem will work.
> >
> > 7.	“Definition of $T$ and replacing $T$ with $Tn$.”
> >
> > We will be more clear about the definition. $T$ is the total number of interactions between two nodes. It can be replaced by $\Omega(T_{parallel}n)$: If we look at the interactions ordered linearly by the time when they occur, we can split them in $T_{parallel}$, consecutive chunks, where each chunk contains $\Omega(n)$ operations and all operations within a chunk happen in parallel. (This transformation to parallel time is standard in gossip and population models.)
> >
> > *We have added a clarification discussion on this point.*
> >
> > B. Further Questions.
> >
> > 1.	“Merging section I with theorem 4.1.”
> > We merged section I with theorem 4.2 since , Theorem 4.1 uses second moment bound and $\rho$ does not appear.
> > In the case of theorem 4.2 , the reviewer is correct: there is a term with $\rho^2$ (we replace $\sigma^2$ with $\sigma^2+4\rho^2$).
> >
> > *Thank you for this nice suggestion, a version of which we implemented in the revision.*
> >
> > 2.	“Lemma F.3 is confusing, $\Gamma(t)$ should decrease with $t$.”
> >
> > The purpose of lemma F.3 is to show that local models of nodes do not diverge, as $t$ increases. For this, it is not required that $\Gamma(t)$ decreases with $t$. We could indeed use diminishing step sizes, but it would not necessarily improve the convergence bound and it also would cause additional overhead of coordinating step sizes between the nodes (We would have to make sure that step sizes of nodes do not differ by too much).
> >
> >
> > 3.	“Can you also show the run time plot for ResNet?”
> >
> > *Yes, we have added this to the revision.*
> >
> > C. Optional improvements.
> >
> > [*We have clarified all these points.*]
> >
> > We thank the reviewer for the provided suggestions. We will address them as follows:
> >
> > 1.	“$\frac{r}{\lambda_2} \ge 1$.”
> >
> > We would like to point out that in the case of the fully connected graph $r=n-1$.
> > and $\lambda_2=n$, we will add this example to the discussion.
> >
> > 2.	“$\tilde h_i^s$ depends on $\tilde g_i$.”
> >
> > We apologise for not being more precise, we skipped superscript in the case of
> > $\tilde g_i$, This would make $\tilde h_i^s$ to depend on $\tilde g_i^0, \tilde g_i^1, …, \tilde g_i^{s-1}$ .
> >
> > 3.	“Extra ‘-’.”
> >
> > Thank you for pointing this out, we will correct it.
> >
> > 4.	“Missing $(n-2)/n$ factor.”
> >
> > $(n-2)/n$ factor in front of the dot product disappears because of the observation before equation (18) ($(n-2)/n$ becomes 1, since $-2/n$ contributes towards the term which is equal to 0).

---

### Official Review · AnonReviewer4 · 2020-10-28
**simple and effective distributed SGD, with asynchronous, decentralized and reduced communication extensions**

**Rating:** 7
**Confidence:** 3

**Review:**

### Summary
The paper proposes and analyses a distributed learning algorithm for training with Stochastic Gradient Descent a global model on a regular graph, that allows for local and asynchronous gradient updates. Nodes continuously update their local models $X^i$ by gradient descent, while they communicate with their peers (a peer at a time) and update their local model with the pair model average $\frac{X^i + X^j}{2}$. Three extensions of the algorithm are also proposed to relax different constraints, while maintaining the convergence guarantees:
1. synchronous updates and decentralized data: if the number of local gradient updates $H_i$ before an edge update is constant, convergence guarantees hold for decentralized data, as long as partitions are i.i.d. from the original distribution;
1. asynchronous updates: the number of local gradient updates $H_i$ can vary between nodes and between every edge update;
3. reduced communication: model exchanges can be quantized to reduce communication complexity.
Experiments in the distributed setting are carried out for image classification and speech recognition, showing that the algorithm is generally able to achieve performance comparable to a model trained in the centralized setting at increased execution time, but faster than state-of-the-art distributed SGD methods.

### Significance and clarity
Contributions are significant and novel, to the best of my knowledge. They consider several settings, which are all theoretically founded. However, the paper is generally hard to follow, also because it has many contributions that are cited in the main text but deferred to the appendix. Still, it would help to clarify the following points from the beginning:
1. what the authors mean by decentralized, later explained as decentralized model updates, but centralized/distributed data for the experiments;
2. define $T$ (global number of edge updates) and $H$ (number of local updates in between edge communication);
3. where and when quantization is applied and why it helps in reducing communication complexity in the main text.

### Remarks on theoretical analysis
Theorem 4.1 shows that the average second moment of the loss gradient evaluated at the average model $\mu_t$ is bounded and decreases with $T$, proving that the model updates converge to a local minimum. This bound however stands for the average of all models obtained at each global step $t$, meaning that it is not necessarily a tight bound for the second moment of the last obtained model, which is the bound we are ultimately interested in.
It would be also interesting to report communication complexities, with and without quantization, and compare them to state-of-the-art methods.

### UPDATE
I thank the authors for addressing my concerns and confirm my initial rating.

---

> ### Author Response · Authors · 2020-11-14
> **Individual response**
>
> [*This response was slightly updated to reflect the contents of the revision.*]
>
> Thank you for your feedback! We address the main issues below, in order:
>
> A.	Points on significance and clarity
>
> We thank the reviewer for these suggestions, which we will address carefully in the revision. In short:
> 1.	“meaning of decentralized.”
>
> Indeed, we mean decentralized model updates, in the sense that each node has its own (possibly different) version of the model.  *We have clarified this in the introduction and throughout the paper.*
>
> 2.	We will indeed define $T$ and $H$ at the very beginning.
>
> 3.	“where and when quantization is applied.”
>
> Whenever nodes $i$ and $j$ communicate at step $t$ , instead of model $X_t^i$ node         $j$ receives a quantized version of it (or alternatively reads quantized version from the shared buffer to which $i$ wrote a quantized version earlier). We provide the full algorithm in Appendix G.
>
> *We dedicated a paragraph to explaining the impact of quantization on communication cost, and the relationship to prior work.*
>
> B.	Remarks on theoretical analysis
>
> 1.	“Second moment of the last obtained model.”
>
> This is a great question, but unfortunately it is not known whether one can provide such a bound on the second moment of the last model even in the classical case of non-convex, single-machine SGD. The guarantees we provide are standard in this setting (see for instance the work of [Lian et al., 2017, 2018] as referenced in our paper).
>
> 2.   Communication complexity:
>
> *We provided an accounting of total communication complexity with and without quantization in the revision. This is in the discussion of the main theorem, as well as in the discussion of the quantization extension.*

---

### Author Response · Authors · 2020-11-24
**Revision Submitted**

We thank the reviewers again for their feedback.

We submitted a significant revision, which addresses all the reviewer comments as per our individual replies. In the PDF version, the significant changes are marked in blue. We will continue to make minor revisions until the deadline.

The major changes are the following:

* We unified the variance-based analysis extension (now Extension 1) to also allow for non-i.i.d. data, as part of the same argument. This is a significant technical improvement in the revision.
* We discuss the algorithm's communication complexity, with and without quantization being applied.
* We explained the role of the $T \geq n^4$ requirement and clarified that it significantly improves upon prior work on decentralized algorithms.
* We motivated the choice of graph topology via citations to the literature on supercomputing and cloud networks.
* We clarified the role of the quantization scheme of Davies et al. in our quantized algorithm, and why standard (unbiased) quantizers wouldn't work.
* We added running times vs. accuracy for ResNet18/ImageNet, and a breakdown of average time/batch vs node count for the various schemes.
* We re-wrote part of the introduction for concision and clarity.
* We made sure all notation is clear before it is used.

---

### Decision · Program_Chairs · 2021-01-07
**Final Decision**

**Decision:**

Reject

**Comment:**

The reviews were a bit mixed: on one hand, by combining and adapting existing techniques the authors obtained some interesting new results that seem to complement existing ones; on the other hand, there is some concern on the novelty and on the interpretation of the obtained results. Upon independent reading, the AC agrees with the reviewers that this paper's presentation can use some polishing. (The revision that the authors prepared has addressed some concerns and improved a lot compared to the original submission.) Overall, the analysis is interesting but the significance and novelty of this work require further elaboration. In the end, the PCs and AC agreed that this work is not ready for publication at ICLR yet. Please do not take this decision as an under-appreciation of your work. Rather, please use this opportunity to consider further polishing your draft according to the reviews. It is our belief that with proper revision this work can certainly be a useful addition to the field.

Some of the critical reviews are recalled below to assist the authors' revision:

(a) The result in Theorem 4.1 needs to be contrasted with a single machine setting: do we improve the convergence rate in terms of T here? do we improve the constants in terms of L and M here? What is the advantage one can read off from Theorem 4.1, compared to a single machine implementation? How should we interpret the dependence of (optimal) H on r and lambda_2?

(b) The justification for $T \geq n^4$ is a bit  weak and requires more thoughts: one applies distributed SGD because n is large. What happens if T does not satisfy this condition in practice, as in the experiments?

(c) Extension 1 perhaps should be more detailed as its setting is much more realistic than Theorem 1. One could use Theorem 1 to motivate and explain some high level ideas but the focus should be on Extension 1-3. In extension 2, the final bound seems to be exactly the same as in Theorem 1, except a new condition on T. Any explanations? Why asynchronous updates only require a larger number of interactions but retain the same bound? These explanations would make the obtained theoretical results more accessible and easier to interpret.